# Unravelling the mystery of "Madagascar copal": Age, origin and preservation of a Recent resin

**Xavier Delclòs**[1]*, **Enrique Peñalver**[2], **Voajanahary Ranaivosoa**[3], **Mónica M. Solórzano-Kraemer**[4]

**1** Departament Dinàmica de la Terra i de l'Oceà, Facultat de Ciències de la Terra, Institut de Recerca de la Biodiversitat (IRBio), Universitat de Barcelona, Barcelona, Spain, **2** Instituto Geológico y Minero de España (Museo Geominero), Valencia, Spain, **3** Département Bassins sédimentaires, Evolution et Conservation, Faculté des Sciences, Université d'Antananarivo, Antananarivo, Madagascar, **4** Senckenberg Research Institute, Frankfurt am Main, Germany

* xdelclos@ub.edu

**Data Availability Statement:** All relevant data are within the paper.

**Funding:** Our work has been financially supported by • Spanish Ministry of Economy, Industry and

## Abstract

The loss of biodiversity during the Anthropocene is a constant topic of discussion, especially in the top biodiversity hotspots, such as Madagascar. In this regard, the study of preserved organisms through time, like those included in "Madagascar copal", is of relevance. "Madagascar copal" originated from the leguminous tree *Hymenaea verrucosa*, which produced and produces resin abundantly. In the last 20 years, interest has focused on the scientific study of its biological inclusions, mainly arthropods, described in dozens of publications. The age and origin of the deposits of "Madagascar copal" have not yet been resolved. Our objectives are to determine its age and geographical origin, and thus increase its scientific value as a source of biological/palaeobiological information. Although *Hymenaea* was established in Madagascar during the Miocene, we did not find geological deposits of copal or amber in the island. It is plausible that the evolution of those deposits was negatively conditioned by the type of soil, by the climate, and by the development of soil/litter microorganisms, which inhibit preservation of the resin pieces in the litter and subsoil over 300 years. Our results indicate that "Madagascar copal" is a Recent resin, up to a few hundred years old, that originated from *Hymenaea* trees growing in the lowland coastal forests, one of the most endangered ecosystems in the world. The included and preserved biota is representative of that ecosystem today and during historical times. Inclusions in this Recent resin do not have the palaeontological significance that has been mistakenly attributed to them, but they do have relevant implications for studies regarding Anthropocene biodiversity loss in this hottest hotspot.

## Introduction

Madagascar is an island in Eastern Africa located in the Indian Ocean (Fig 1) between 11˚57 S and 25˚29 S and is considered one of the few hottest hotspots of biodiversity in the world [1],

Competitiveness (Projects CGL2014- 52163 and AEI/FEDER, UE CGL2017-84419) • National Geographic Global Exploration Fund Northern Europe (Project GEFNE 127- 14) • German VolkswagenStiftung (Project N. 90946).

**Competing interests:** The authors have declared that no competing interests exist.

especially in relation to the tropical evergreen rainforest, which extends parallel to the eastern margin of the island. For more than 80 million years, Madagascar has been geographically isolated from the African continent, which is why palaeontological studies of the biota are of great relevance [2]. Nevertheless, it is postulated that a land bridge connected Africa and Madagascar between 45 and 26 Ma ago [3]. During the Miocene the continent of Africa was extensively covered by tropical rainforests [4], however, the presence of palaeosoils and dry fossil-floras seems partly to contradict this idea [5].

The biota embedded in resins, sub-fossil resins, and ambers are preserved three-dimensionally with fine detail in external and some internal characters, sometimes including soft-parts,

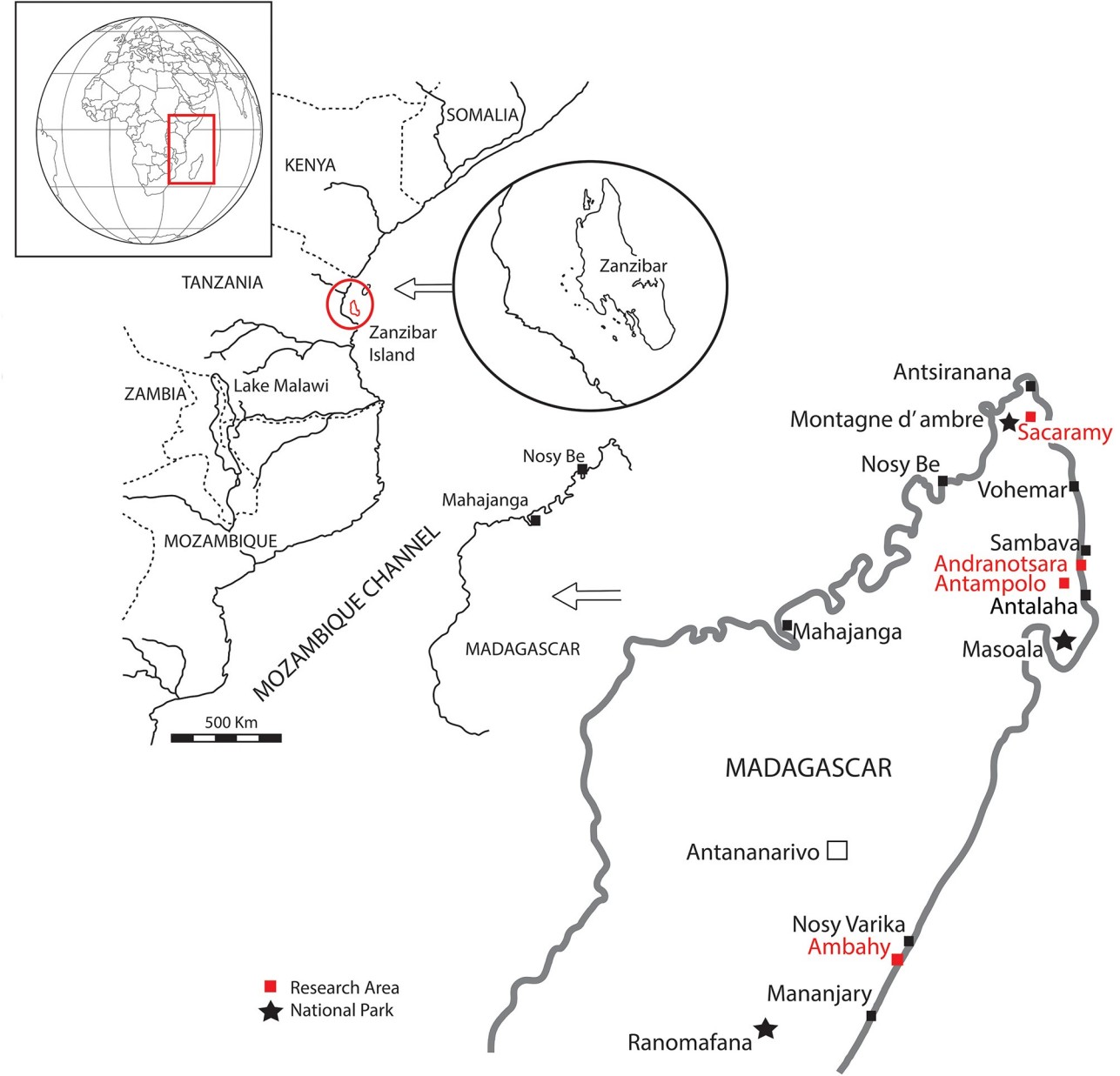

**Fig 1. Map showing the countries that surround the Mozambique Channel, with the localities cited in the text.**

providing one of the best-preserved arthropod assemblages and valuable palaeobiological data. It has been argued that the study of inclusions in copal is important for gaining knowledge of the diversity that could be extinct or undiscovered [6,7]. Copal usually contains extant, undescribed species, so it can hardly be expected to increase knowledge of the evolution of insects. However, it may be crucial for the detection of changes during the Anthropocene in resin-producing forest ecosystems, principally in relation to the current rapid, human induced, modification of these terrestrial environments.

The age of "Madagascar copal" has not been accurately identified, mainly because of problems in finding and studying the geology of possible outcrops. However, in the last twenty-four years, several articles on arthropods included in "Madagascar copal" have been published [8–11]. For all of them the age and the geographical origin of the studied samples were imprecise or unavailable. The age attributed to "Madagascar copal" has varied within a broad range of between tens of years and a few million years [12–14].

The research carried out in Madagascar by us during the last six years, which focused on the leguminous tree *Hymenaea verrucosa* Gaertner, 1791[15], led us to investigate two questions that we consider essential for the development of biological and palaeobiological investigation of bioinclusions contained in resins: 1) How old is the so-called "Madagascar copal", about which a large list of taxonomic works on its bioinclusions have been published?, 2) What is its geographical origin?

In palaeontological and biological studies, the precise location of the collection site and its dating are mandatory. Of particular concern is the number of published works about the biota included in "Madagascar copal" that do not provide accurate data about the age or place of origin of the samples studied, which hinders the development of further works. Some species studied within "Madagascar copal" had already been found existing today [10,16–18] but others are only known from copal [11,19]. Our study indicates that the nature of "Madagascar copal" has long been misinterpreted and better explains the presence of modern taxa trapped inside.

## Historical background

### Copal as a scientific term

The scientific term "copal" is ambiguous, mainly because it is used without consideration of the typology of resins based on their different states of polymerization. It is used across commercial and scientific fields, such as geology, geochemistry or palaeontology, creating great confusion. Copal is considered an intermediate stage between resin and amber or a tree resin that is not completely fossilised (polymerised), but no specific age has been assigned to it. The term copal is derived from the word *copalli* (in Aztec *náhuatl*) meaning "resin" used as incense. Major pre-Hispanic cultures used it to refer to resins produced by different tree genera such as *Protium* Burman, 1768 and *Bursera* von Jacquin ex Linnaeus, 1762 (Burseraceae), *Pinus* Linnaeus, 1753 (Pinaceae) and *Liquidambar* Linnaeus, 1753 (Altingiaceae). The Spanish settlers arriving in the Americas during the 15th –16th centuries, also applied the term copal to resins of different trees used by the Aztecs [20], and since then, up to today, the term has been used broadly for all non-fossilised resins worldwide.

The classification of resin or amber is based on the ensemble of chemical compounds it contains [21]. The classification can be more direct if the amber or resin can be associated to a fossilized or living tree. Neither copal nor amber has a crystalline structure, but as tree products both are organic materials. Polymerisation of organic hydrocarbon molecules changes the resin to sub-fossil resin, and a cross-linkage reaction between chains of these hydrocarbons

produces amber [22]. Copal and amber can be differentiated using FTIR spectroscopy by observing precise exocyclic methylene bands [23].

The age of the sedimentary deposit is a criterion used to consider a subsoil resin as copal and is often limited to between 10,000 years and 5 million years [12,24]. Schlüter and Glöckner [25] considered that a one-million-year-old resin is to be called amber, when it is younger, copal. Anderson ([26], p. 252) proposed an objective scale based on [14]C dating, stating that "*fresh resins and ambers are members of a continuous series*". For Anderson, resin under 250 years old should be named "modern resin", while resins between 250 years old and 5,000 years old should be named "ancient resin" and resins aged between 5,000 years and 40,000 years, "fossil resins". He did not consider using the term copal for any of the temporal stages of the development from resin to amber. Langenheim ([20], p. 146) considered the term "copal" controversial but generated confusion when she mentioned her intention to follow the terminology proposed by Anderson (op. cit.), and in later chapters used the expression "hard copals" without giving any indication of having applied Anderson's terminology (p. 392). She even used the term "copal" to refer also to the resin found in the fruit pods of *H. verrucosa* (p. 397). Kimura et al. [27,28] considered that the polymerisation that transforms copal into amber proceeds very slowly. They proposed that it takes about 13 million years to reach the halfway point; this means that copal requires some millions of years to mature into amber. Vávra ([29], p. 215) considered copal as "*Resins having not yet undergone all the steps of fossilization like polymerisation and maturation*". Lambert et al. [30] considered that resin that is thousands of years old, up to a million years old better retains its original molecular structure and is more aptly termed "copal".

Thermal analysis allows an estimation of the age of resins and ambers [31]. In this way "Madagascar copal" has been dated from Holocene to Recent (1,000–100 a) [32]. However, as of today there is no technique using the loss of volatile components as an age indicator for fossil or sub-fossil resins, probably because of the complexity of these components [33]. It must be remembered that there is a gap of 13 to 15 million years between the youngest amber deposits dating from the Middle Miocene (New Zealand, Peruvian, Mexican, and Dominican amber) and the oldest copal deposits, which are a few thousand years old (New Zealand, Colombian or Dominican copal).

## Copal as a trade product in East Africa

Madagascar houses several angiosperm resin-producing trees, mainly in the east of the island in the evergreen rainforests such as *Canarium* spp. (Burseraceae), and in the lowland coastal forest such as *Hymenaea verrucosa* (Leguminosae) (Fig 2). The resin produced by *Canarium* spp. was historically called "elemi" or "alemi" by merchants, and "copal" was the name used for the resin produced by *H. verrucosa* in the context of East African trade [34].

"East African copal" was exported from about 2,500 BCE [35–38]. It has commonly been found in archaeological sites in East Africa dating from the Early Iron Age (2nd–3rd centuries BCE) [39,40], and was traded intensely from the 11th century [34,36] to the beginning of the 20th century between parts of eastern Africa and diverse countries. This resin was traded for thousands of years, together with other resinous products such as gum arabic (Fabaceae: *Acacia* Miller, 1754), myrrh (Burseraceae: *Commiphora* von Jacquin, 1797) and frankincense (Burseraceae: *Boswellia* Roxburgh ex Colebrooke, 1807) [34,35]. All these resins, or oleoresins, and copal can be differentiated by their diterpenic or triterpenic geochemical characters [41]. "East African copal" is a diterpenic resin containing ozic acid, an enantiomer of communic acid that polymerises as soon as it is exuded from the tree and comes into contact with air and light [42].

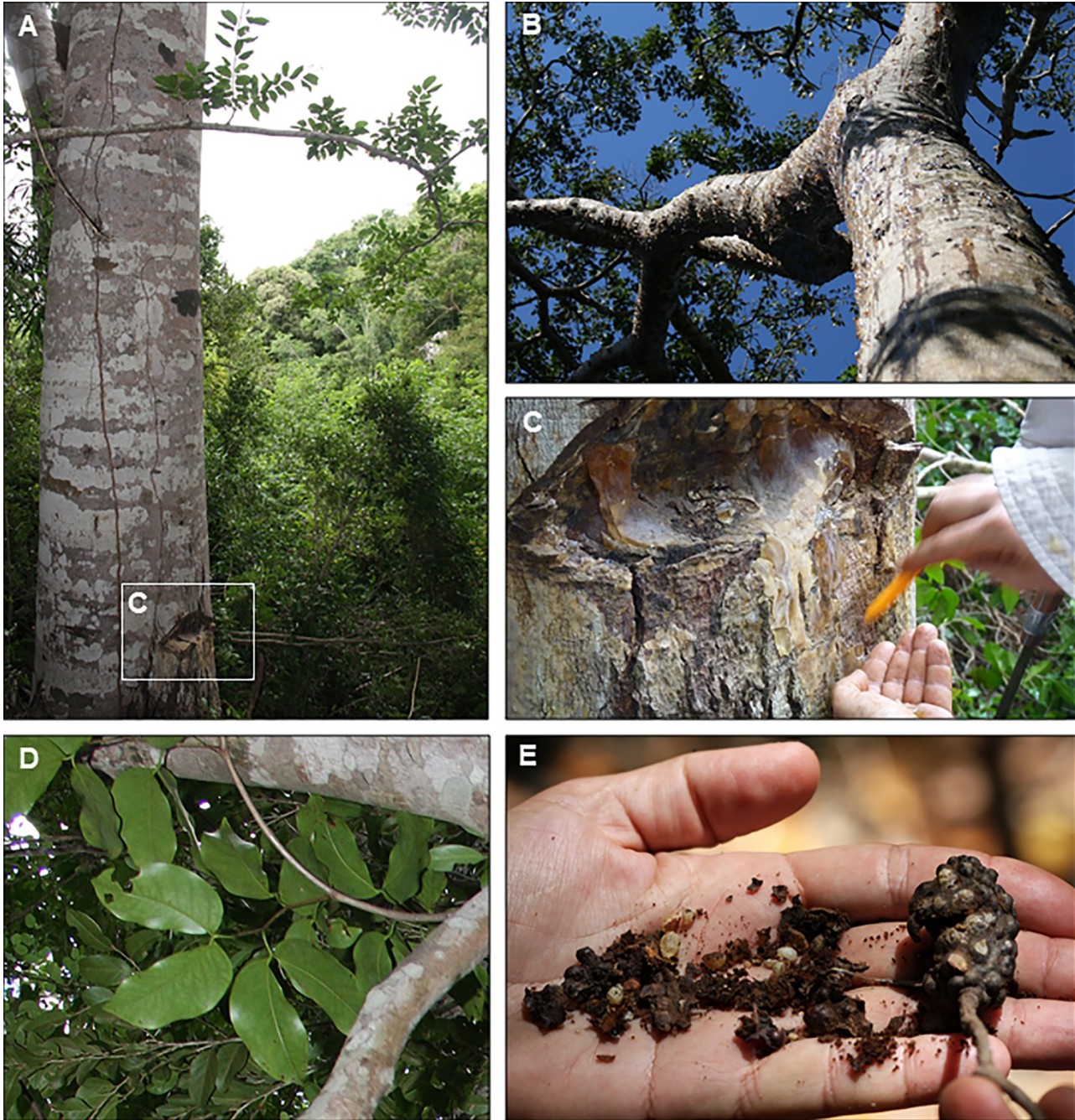

**Fig 2. Hymenaea verrucosa.** Madagascar: (A) Trunk, in Sacaramy (Antsiranana), with a significant injury at its base. (B) Habitus of an individual, producing abundant resin in several parts, Andranotsara (Sambava). (C) Detail of the injury in A, with important resin production. (D) Leaves, Ambahy (Mananjary) (leaflet length ca. 6.5 cm). (E) Decomposed fruit showing seeds and pod resin beads (these developed as resin pockets in the pod wall) and intact fruit 4 cm in length, Ambahy (Mananjary). Observe the characteristic roughness of the external surface produced by these resin pockets, which gives the species its name.

The oldest artefacts made with "copal" from *Hymenaea verrucosa* from East Africa date from 2,500 to 2,400 BC and were found in modern-day Iraq [35], and the oldest pieces with bioinclusions (found in manufactured beads) date back to 2,000 BC and were found in Egypt [43].

The "copal" occurring in countries on both sides of the Mozambique Channel (mainly Tanzania, Mozambique and Madagascar) has regularly been traded as the "East African copal" or the "Zanzibar copal". This was because the Sultanate of Zanzibar (today Tanzania) was, for centuries, the central hub of commercial activities between Africa and the other harbours in the Indian Ocean, such as the Malagasy ports [44]. In the 16[th] century, Zanzibar began to trade "copal" with Europe under the denomination "gum copal" or "Zanzibar copal". Although trade between Madagascar and the east coast of Africa possibly began in the 7[th] century, there is no documentation suggesting that "Madagascar copal" trade began before the mid-18[th] century [45]. Possibly the first reference to "Madagascar copal" as an exportable item of merchandise is from 1767, when Valigny (in [46]) reported a Malagasy resin in China. At the end of the 18[th] century "copal" was exported in large quantities from the harbours on the east coast of Madagascar [47] to Europe, the United States of America, and some south-eastern Asian countries such as India and China, for use in the manufacture of varnishes. However, the "copal" trade in Madagascar did not develop widely, unlike in Tanzania, because the Malagasy were not careful enough in their preparation of it, which greatly reduced the price [48].

## "Madagascar copal", geographical origin and age

Elton [49] and McIntosh [50] described the extraction of the "Zanzibar copal" in Tanganyika from the subsoil, at a depth of below 120 cm, and yet very few diggers appear to go beyond 90 cm deep. According to Rice [51], the "Zanzibar copal" was found by natives at between 60 cm to 120 cm underground over a wide area on the coast opposite the island of Zanzibar.

The first reliable reference to "Madagascar copal" as a trade name came from Filhol [52]. For Filhol, "Madagascar copal" was hard copal traded as large flat, yellow pieces to India. "Madagascar copal" was collected from the soil and, according to Howes ([53], p. 98), "*Madagascar copal resembles that from East Africa in hardness and other characters but does not show the characteristic "gooseskin". It is collected from the soil and may be present in large deposits mainly in the north part of the island. This copal is considered to be derived from the same botanical source as the East African. Resin may also be obtained by tapping*" [54]. In 1868, Oliver said ([55], p. 71) that "*Copal gum might be collected in Madagascar to almost any extent...there was but little collected by the natives*". During the second half of the 19[th] century, officers and soldiers of garrisons located on the Malagasy coast forced the people of the surrounding towns to sell them part of their produce at low prices, which they then resold to the merchants, or they sent their slaves to collect both bees-wax and "copal" [56]. Nevertheless, the "copal" harvested was only supplementary to rubber collection and trade [50]; the "copal" was not, in itself sufficiently remunerative to harvesters, but it served to compensate for the journey involved in bringing their loads to the coast. The whole of this trade was controlled by Indians and Arabs who were established at Vohemar and Antsiranana. At the end of the 19[th] century, "copal" was also harvested and sold as merchandise in the region of Tamatave [57].

All these ancient references suggest that "Madagascar copal" was obtained in the northern and eastern coastal regions of Madagascar (Fig 1), where the resin-producing tree *Hymenaea verrucosa* lives today (Fig 2). Also, Rakotovao [58] proposed the area of Antsiranana as the source of the "copal" and dated it as Quaternary or Holocene. Previous authors [59–61] identified the species *H. verrucosa* as the producer tree of "Madagascar copal". However, comparisons with other large resin producers in Madagascar, such as *Canarium* Linnaeus, 1759

(Bursaceae), which is abundant in the evergreen rainforests (such as at *Montagne d'Ambre*, Masoala or Marojejy in Sambava), have not been made.

The presence of *Hymenaea* in Madagascar needs to be explained. Phylogenetic analysis by Fougère-Danezan [62] indicated that *Hymenaea* started in continental Africa, as previously suggested by Langenheim and Lee [60], along with its sister genus *Guibourtia* Bennett, 1857, which today grows in western African forests. Fougère-Danezan [62] suggested that the clade [*Hymenaea* + *Guibourtia*] originated in Africa 25 to 19 Ma ago. This has been corroborated by the discovery of the "amber of Ethiopia", initially dated from the Cenomanian [63] but after re-evaluation by Perrichot et al. [64,65], as Miocene and produced by the genus *Hymenaea* [33], and phylogenetically reassess by De la Estrella et al. [66]. Later, the uplift of the East African Plateau during the Miocene, about 13.5 Ma ago [67], favoured the development of *Guibourtia* in the occidental margin with its more humid environments, and *Hymenaea* in the Eastern lowlands with better adaptation to annual periods of water scarcity. The East African Plateau limited westward expansion of *Hymenaea* [68] and favoured its dispersion to several Indian Ocean islands. Zoochory is not reported as a dispersal mechanism for *H. verrucosa* in East Africa and Madagascar, nevertheless, the fauna, mainly rodents and other mammals, can disperse and transport fruits and seeds of Neotropical *Hymenaea* species [69]. Hydrochory is the main mechanism for seed dispersal of the buoyant fruits. Thus, dispersion of *Hymenaea* possibly occurred via short distance dispersal across the Mozambique Channel by birds and/or marine currents. However, during the Upper Miocene and up to today, marine currents were and are largely unfavourable for marine dispersal from Africa to Madagascar [70–72]. Madagascar is surrounded by several important oceanic currents [73,74]. *Hymenaea verrucosa* possibly disseminated from East Africa to Madagascar and surrounding islands during the Early Miocene, mainly during hurricane seasons [70,75]. This species is restricted to lowland coastal forests; normally water-dispersed seeds germinate near the water through which they have been dispersed.

Because the presence of *Hymenaea* in Madagascar goes back to the Miocene it would be conceivable that deposits, not only of copal but also of amber, could be found. However, Meunier [76,77], based on the study of its inclusions, had doubts about such an old age for East African copal. Williamson [78] also considered "Madagascar copal" of Recent origin. Hills [79] suggested that the study of bioinclusions permits the differentiation of amber from copal and considered that amber includes mainly extinct insect species, while copal includes mainly extant insect species. Nevertheless, he did not differentiate between copal and Recent resin. According to Vankerkhoven et al. [80], the age of "Madagascar copal" is uncertain and varies with the depth of mining, but the fauna included in the copal pieces differs from Recent fauna, which suggests that it was, at least partly formed several thousand years ago. However, Vankerkhoven's statement is not correct because: i) there is no evidence that the Malagasy people obtained "Madagascar copal" from underground sites; ii) the studied pieces do not have a proper geographic register, and iii) The Recent arthropod fauna from the hotspot of Madagascar is thus far inadequately known. Poinar [12] considered that the existence of copal in Madagascar dated from between 3 and 4 Ma ago, while Le Gall et al. [81] assigned an age of between 1 and 2.6 Ma, in both cases without evidence. Other authors [13,29,82–84], suggested that "Madagascar copal" from the north coast (Antsiranana) had a presumed age of between 10,000 and 50,000 years; in no case was the exact provenance of the analysed pieces indicated.

## Material and methods

In a new approach aimed at determining actual age, we systematically constructed two pits and subsequently applied radiocarbon dating analyses ($^{14}$C) to the resin samples obtained at

different depths in two different *Hymenaea* forests, as well as to some pieces housed in museums and others obtained in local markets. To determine the potential localities where "Madagascar copal" was obtained, either by excavation in the soil or maybe by harvesting fresh resin from the *H. verrucosa* trees, we first conducted an intensive bibliographical search that allowed us to delimit three main geographic regions; Mananjary in the south-east, *Montagne d'Ambre* —Antsiranana in the north, and Sambava in the north-east (Fig 1). In these regions, taphonomic and geological studies have been carried out in search of copal-bearing geological deposits.

## Ethics statements

All necessary permits were obtained for the described study, which complied with all relevant regulations. The Ministère de l'Environnement, de l'Écologie et des Forêts gave us permission to work in the Malagasy protected areas (192/13/MEF/SG/DGF/DCB.SAP/SCB; 060/15/ MEEF/SG/DGF/DCB.SAP/SCB, and 192/17/MEEF/SG/DGF/DSAP/SCB.Re).

## Collection localities and environmental characteristics

The search for copal deposits (during 2013, 2015, and 2017), was confined to the three areas that have traditionally been proposed in the literature as the origin of the copal used in taxonomic studies (Fig 1).

1. *Mananjary region* (2013, research area between Nosy Varika and Ambahy). *Hymenaea* is common and lives here under an equatorial climate (Af in the Köppen-Geiger climate classification system; see Kottek et al. [85]). The average annual temperature is 23.1˚C (the variation in annual temperature is around 6.2˚C). The annual rainfall is high, averaging 2,439 mm, with precipitation even during the driest month. We harvested resin directly from the *H. verrucosa* trees [15]. *Hymenaea* trees are used to provide shade for the vanilla plants.

2. *Montagne d'Ambre—Antsiranana region* (2015, research area between Sacaramy and Antsiranana). *Montagne d'Ambre* (Amber Mountain) is usually reported by Malagasy dealers and in some publications [16,86] as the locality where "Madagascar copal" is found. The people there do not know the origin of the names *Montagne d'Ambre* or *Cap d'Ambre*, but in maps from the 16th century (1588, Africae Nova Tabula), *C. del Ambar* (Amber Cape) is a location more to the north of Madagascar. *Montagne d'Ambre* corresponds to an evergreen rainforest that is not conducive to the development of *Hymenaea* forests. Between Antsiranana (= Diego Suarez, close to "*Cap d'Ambre*"), and Sacaramy, *Hymenaea* lives under a tropical savanna climate (Aw in Köppen-Geiger classification). Rainfall is much greater in summer than in winter. The average annual temperature is 26.5˚C; the temperature in December averages 28.0˚C and at an average 24.8˚C, July is the coldest month. The average annual rainfall is 1,156 mm. The driest month is September, with 11 mm of rain. In January, the precipitation reaches its annual peak, with an average of 320 mm.

3. *Sambava region* (2017, research area between Andranotsara and Antampolo). In this region the genus lives under an equatorial climate (Af in Köppen-Geiger classification). The rainfall in Sambava is significant, with precipitation even during the driest month. The temperature averages 24.7˚C; the warmest month is February with an average temperature of 26.8˚C and at an average 22.2˚C, August is the coldest month. The annual average rainfall is 2,030 mm. The driest month is November, with 77 mm of rainfall. In January, the precipitation reaches its annual peak, with an average of 257 mm. In this region the trees are also used to provide shade for the vanilla plants, stems are obtained for charcoal production, and branches and trunks for building. In this area two pits were dug to study the resin

accumulation and burial, as described below. To observe the soil horizons of the Malagasy *Hymenaea* forests and the possible preservation/degradation of resin in it, two test pits were dug, each with an area of 150 x 150 cm (Fig 4). One was dug in Andranotsara (pit Q1), to a depth of 175 cm (GPS: S14 38.472'—E050˚12.467') (Fig 4A and 4B) and a second in Antampolo (pit Q2), to a depth of 140 cm (GPS: S14˚ 43.468'—E050˚12.860') (Fig 4C). The grid was made with nails and rope and excavated with picks and shovels to obtain copal or resin samples from different levels of depth. Two synthetic pedologic logs (Fig 5A and 5B) were drawn (description below, section 4.2), and soil samples were collected from each of the differentiated horizons and sub-horizons (Figs 4D–4G, 8A, 8B and 8E–8G).

## FTIR analyses

Using the facilities of the University of Barcelona's Science and Technology Centres, we conducted Fourier transform infrared spectroscopy (FTIR) analyses of sub-fossil/Recent resins and ambers. We used a diamond PerkinElmer FT-IR Spectrometer Frontier; software version 10.4.2 (2014). Analysed samples are housed at the University of Barcelona (Department of Earth and Ocean Dynamics, Faculty of Earth Sciences) and at Senckenberg Research Institute, Frankfurt, Germany. The analysed samples are summarised in Table 1.

To characterise the original material of the soil beads found abundantly in the pit horizons O and A of the pits dug, FTIR analyses have also been carried out (Fig 5C): one sample of resin

**Table 1. Characteristics of the samples, FTIR analysis.**

| Sample | Tree species | Material (sensu Anderson, 1966) | Age | Collected | Locality | Country | Additional data |
|---|---|---|---|---|---|---|---|
| RNP-RC01 | *Canarium madagascariensis* | resin | Recent | 2013 | Ranomafana National Park | Madagascar | from a *C. madagascariensis* trunk |
| CM01 | Madagascar copal | modern resin | < 300 years | 2013 | Antsirabe market | Madagascar | with *H. verrucosa* flower as bioinclusion |
| CM02 | Madagascar copal | modern resin | < 300 years | 2013 | Antsirabe market | Madagascar | with *H. verrucosa* leaflet as bioinclusion |
| RHvA01 | *Hymenaea verrucosa* | resin | Recent | 2013 | Ambahy (Nosy Varika) | Madagascar | from a *H. verrucosa* branch |
| RHvA02 | *Hymenaea verrucosa* | resin | Recent | 2013 | Ambahy (Nosy Varika) | Madagascar | from a *H.verrucosa* trunk |
| CM05-Q1 | *Hymenaea verrucosa* | modern resin | < 300 years | 2017 | Andranotsara (Sambava) | Madagascar | from Pit Q1—A Horizon, 0–10 cm depth |
| CM06-Q1 | *Hymenaea verrucosa* | modern resin | < 300 years | 2017 | Andranotsara (Sambava) | Madagascar | from Pit Q1—B Horizon, 10–20 cm depth |
| CM03-Q2 | *Hymenaea verrucosa* | modern resin | < 300 years | 2017 | Antampolo (Sambava) | Madagascar | from Pit Q2—A Horizon, 3–10 cm depth |
| CM04-Q2 | *Hymenaea verrucosa* | modern resin | < 300 years | 2017 | Antampolo (Sambava) | Madagascar | from Pit Q2—B Horizon, 10–20 cm depth |
| CCS01 | *Hymenaea* sp. | ancient resin | Anthropocene | 2008 | Santander region | Colombia | with winged termites as bioinclusions |
| AMS-01 | *Hymenaea mexicana*† / *H. allendis*† | amber | Lower Miocene | 2001 | Simojovel de Allende | Mexico | obtained in mine |
| AD-03 | *Hymenaea protera*† | amber | Lower Miocene | 2018 | El Valle—Siete cañadas | Dominican Republic | obtained in mine |
| AD-04 | *Hymenaea protera*† | amber | Lower Miocene | 2018 | El Cabao—San Rafael | Dominican Republic | obtained in mine |
| SMF Be 13212 | *Hymenaea* sp. | amber | Miocene | 2017 | Debre Libanos | Ethiopia | obtained in mine |

of living *H. verrucosa* tree from Antampolo, piece number RHvA; one sample of several soil beads found in the O horizon of the pit Q2 soil, and one sample of several soil beads found in the surface of the soil in another part of Antampolo. This variety of samples has been analysed to permit comparison between them (Fig 3A–3C).

### Radiocarbon dating of "Madagascar copal"

The $^{14}$C analyses were done by the Beta Analytic laboratory, USA, which reported the age results in "pMC" (percent modern carbon) units rather than BP. Beta Analytic reports in the pMC format when the analysed material has more $^{14}$C than the modern (AD 1950) reference standard. We analysed five pieces of "Madagascar copal", summarized in Table 2.

### SEM imaging

The micromorphology of the sectioned soil beads used a FEI INSPECT (5350 NE Dawson Creek Drive Hillsboro, Oregon 97124, USA) Scanning Electron Microscope (SEM) at the National Museum of Natural Sciences (Museo Nacional de Ciencias Naturales) in Madrid, courtesy of the Spanish National Research Council (CSIC).

## Results and discussion

### Resiniferous tree that produced "Madagascar copal"

The FTIR analyses carried out on resin, copal and amber from different areas, chosen for geographically relevance (see section FTIR analysis) confirm that "Madagascar copal" was produced by the fabacean species *Hymenaea verrucosa*. The resiniferous tree species *Canarium madagascariensis* inhabits the *Montagne d'Ambre*, however no copal originated from the genus *Canarium* (Fig 3A). Copal and amber can be differentiated by FTIR spectroscopy by observing the exocyclic methylene bands at 887, 1642 and 3048 cm$^{-1}$ [23]. In the case of copal, the latter two bands are weak but can be observed. The 887 cm$^{-1}$ band is very intense. In the case of ambers these bands are weak or absent [87]. "Madagascar copal" is characterised by the 1700 > 1730 and 1600 cm$^{-1}$ intense bands which correspond to the ester and carboxyl groups, and to the ester group and $>C = C<$, respectively. It is also characterised by the absence of the 1085 cm$^{-1}$ band (alcohol group), the 972 cm$^{-1}$ band (ester group) and the 792 cm$^{-1}$ band (aromatic group) [88] (Fig 3B). According to McCoy et al. [33], "Madagascar copal" can be differentiated from other resins or copal from neotropical *Hymenaea* ssp. due to the presence of four organic compounds: 1) 13-epimanool, 2) caryophyllene, 3) α-curcumene, and 4) biformene; the first three usually play a role in defence against herbivores, the last is as yet unstudied.

The "Zanzibar copal" was first reported to be related to the tree species *Hymenaea verrucosa* by Kirk [59], suggesting that most of the plant remains would belong to the tree that produced the copal, as demonstrated later by Peñalver et al. [89]. In some samples housed in the Kew Museum of Zanzibar, Kirk observed diverse copal samples containing inclusions of leaves and flowers of *Trachylobium hornemannianum*. Kirk did not know at that time that *T. hornemannianum* had been synonymised with *T. verrucosum* a few years earlier [90]. In 1974, the genus *Trachylobium* was considered to be a junior synonym of the genus *Hymenaea*, *Hymenaea verrucosa* Gaertner, 1791 [60].

The genus *Hymenaea* is a leguminous tree belonging to the tribe Detarieae within the subfamily Caesalpinioideae (= Detarieae sensu De la Estrella et al. [66]), with its origin in continental Africa [60,62]. Today, the species *H. verrucosa* grows in the lowland coastal forest of Kenya, Somalia, Tanzania and Mozambique, mainly between Delgado Cape and Ponta

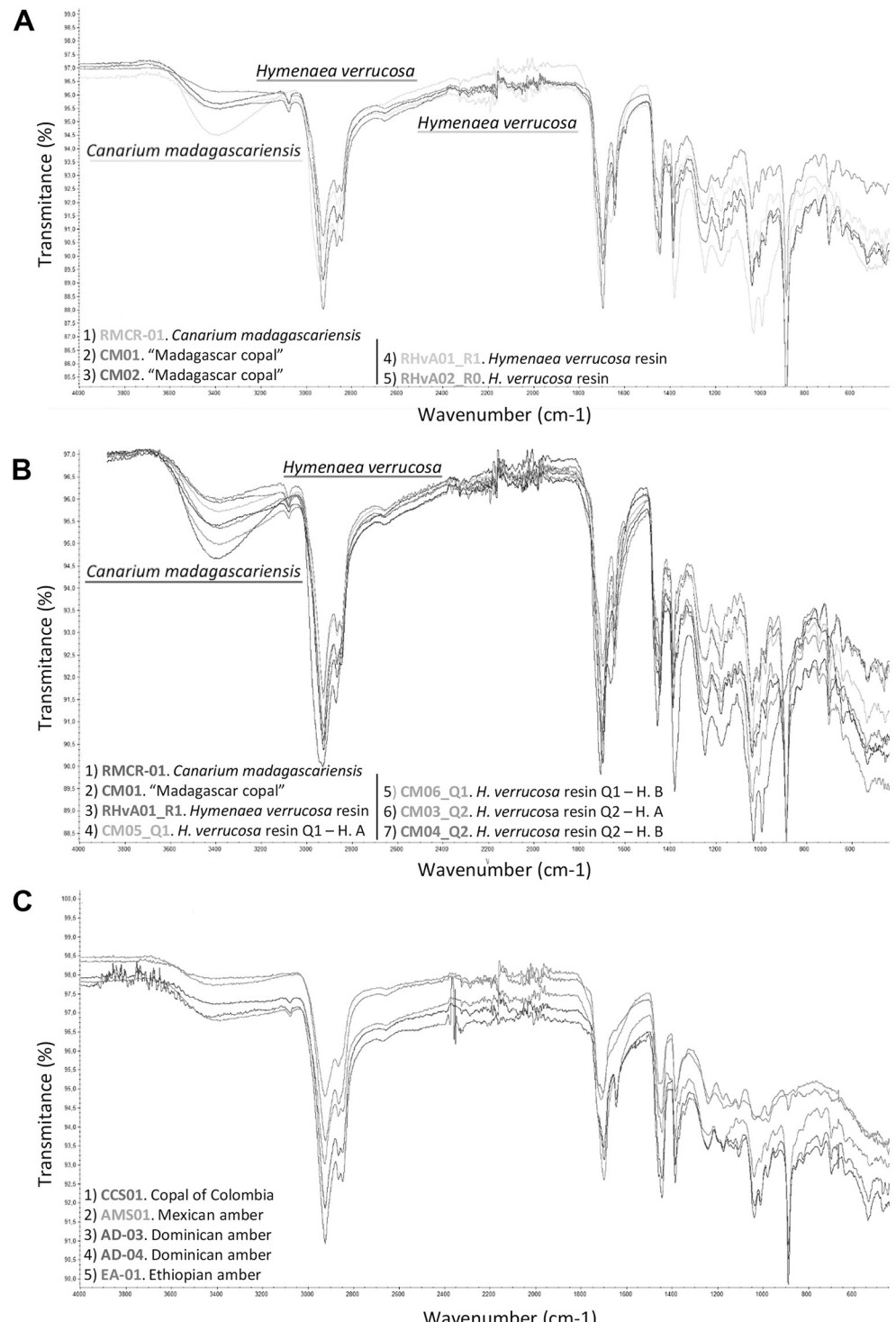

**Fig 3. FTIR analyses comparing the differences between resin, copal and amber produced by *Hymenaea* spp. and *Canarium madagascariensis*.** (A) FTIR analyses of the samples in order to identify the tree-resin producer of "Madagascar copal", include (see Table 1): *Canarium madagascariensis* and two samples of "Madagascar copal" (CM01 and CM02), two samples of resin from *H. verrucosa* branches (RHvA01-R1 and RHvA02-R0). (B) Malagasy *H. verrucosa* analyses: FTIR analyses that show the comparison between the results of A and other resin, copal and amber produced by *Hymenaea* ssp. The analyses include samples of *Canarium madagascariensis* (RMCR-01), "Madagascar copal" (CM01), resin from *H. verrucosa* branch, (RHvA01-R1), resin from *H. verrucosa* of the Andranotsara pit Q1, found in A horizon A (CM05-Q1) and in the sub-horizon B₁ (CM06-Q1), and resin pieces of *H. verrucosa* of the

Antampolo pit Q2, found in A horizon (CM03-Q2) and in sub-horizon $B_1$ (CM4-Q2). (C) Neotropical *Hymenaea* spp. resin and amber analyses: resin from "copal of Colombia", (CCS01), Miocene Mexican amber (AMS01), Miocene Dominican amber (AD-03) and (AD-04), and Miocene Ethiopian amber (EA-01). Diterpenic resin/copal has some characteristic vibrational group frequencies: characteristic is a low intensity of absorption band at 3080 cm$^{-1}$ that is absent from triterpenoid resin/copal and that corresponds to ν (= C-H), intensity absorption band at 2937–2929 cm$^{-1}$ corresponds to ν$_{as}$(C-H), CH$_3$, CH$_2$ (methylene group), intensity band at 2874–2844 cm$^{-1}$ corresponds to ν$_s$ (C-H), CH$_3$, CH$_2$ (methyl group), intensity bands at 1718 cm$^{-1}$, 1694 cm$^{-1}$, and 1644 cm$^{-1}$ correspond to ν (C = O), intensity band at 1446 cm$^{-1}$ corresponds to δ$_{as}$ (CH$_3$), intensity band at 1386 cm$^{-1}$ corresponds to δ$_s$ (CH$_3$), and intensity band at 888 cm$^{-1}$ corresponds out of plane δ (CH$_2$) of the exomethylene functionality C$_8$-C$_{20}$. "Madagascar copal" and "East African Copal" can be differentiated from "Western African Copal" by the linear slope of the spectra in the case of the resin/copal of West Africa and the intensity of 3411–3422 cm$^{-1}$ that corresponds to ν (OH) of the East African copal. It is possible to differentiate between amber and copal by observing the exocyclic methylene bands at 3048, 1642 and 887 cm$^{-1}$. In the case of copal, the first two bands are not intense, but they are clearly observed, and the band of 887 cm$^{-1}$ is very intense. In the case of ambers, the bands are absent or of very weak intensity.

Gomeni near Maputo [54], and in the islands of Madagascar, Zanzibar, and Mauritius [20,82]. It is the only species of the genus living in Africa and is considered a sister species of the 14 others with Neotropical distribution [20,75,91,92], that were involved in the formation of important Miocene amber and copal deposits [33,93,94]. Resin production is abundant among the tribe Detariae, that includes more than half of the genera such as *Copaifera*, *Daniellia* and *Hymenaea* produced the so-called "African Caesalpinioideae (= Detarioideae) species [61,62,66]. Several Detarieae tree copal", but only the last of these grew in ancient Madagascar; no fossil remains of *Copaifera* or *Daniellia* are recorded on the island.

Coastal forests are the ideal environment for the development of legumes such as *Hymenaea* due to the occurrence of seasonal dry periods. The high nitrogen metabolism of legumes confers a competitive edge in colonising seasonally dry environments, because leaves can be produced/reduced accordingly under unpredictable climate conditions [95,96]. Coastal forests are characterised by distinctive biota because of their development under a climate with a dry season lasting several months [68,97,98]. This situation conditioned the type of biota preserved in resin or copal [15], not evergreen rain tropical forest biota but coastal or lowland/coastal forest biota. These coastal forests are common in eastern Madagascar; plants are usually rooted in sand bordering lagoons and marshes and exposed to salty air washing and hurricane injuries. The coastal forest harbours a very unusual arthropod community [99–102]. This forest degrades rapidly because it occurs in one of the most populated areas. At the same time, it corresponds to one of the least studied environments because research teams focus their studies

**Table 2. Characteristics of the samples and results.**

| Sample | pMC | BP | Cal. AD/ BP | Additional data |
|---|---|---|---|---|
| "Madagascar copal" | 105.20 ± 0.3 | | | Two pieces of "Madagascar copal" radiocarbon dated in June 2012 by Dr Thomas Hörnschemeyer. |
| Andranotsara, close to pit Q1 collected at 50 cm, SMF Be 13311 | | 300 ± 30 | | Resin collected in a hole made by native people close to the Andranotsara pit Q1, in which resin was not found deeper than 50 cm. |
| "Zanzibar copal", SMF Be 3724 | | 80 ± 30 | Cal AD 1710 to 1720 / Cal BP 240 to 230 | The analysed piece arrived at the Senckenberg Research Institute in Frankfurt between 1874 and 1901. |
| Andranotsara pit Q1 at 15–30 cm depth | 140.47 ± 0.52 | | | Fig 8E |
| Antampolo pit Q2 at 20–30 cm depth | | 180 ± 30 | | Fig 8F |
| Antampolo pit Q2 at 30–70 cm depth | 180.83 ± 0.41 | | | Fig 8G |

[14]C analysis. The last five samples were analysed by Beta Analytics. A portion of each of these pieces is housed in the University of Barcelona (Faculty of Earth Sciences).

on rainforests, with greater biodiversity. Arthropod species that live in the lowland forests are less present in Malagasy entomological treatises.

*Hymenaea verrucosa* lives in Madagascar under several climatic regions (*sensu* Köppen climatic classification): in the southeast (Mananjary region) under an equatorial climate, in the northeast (Sambava region) under a monsoon climate, and in the north (Antsiranana region) under a tropical savanna climate. All these areas are included in one of the three most endangered Malagasy ecosystems. From 2001 to 2017, Madagascar lost 3.27 Mha of tree cover, equivalent to a 19% decrease since 2000 (Global Forest Watch [103]). The east coast littoral forest has shrunk by over 90% of its former distribution and now ranks among the most critically threatened ecosystems of the world [104,105].

## Pit excavations and types of soil

During our collection trips from 2013 to 2017 in the south-eastern, northern and north-eastern areas of Madagascar, copal deposits were intensively sought (see section collection localities, for more information). Surprisingly, buried resin pieces were only found in the below the ground in extant *Hymenaea* forests.

Ninety percent of Madagascar's total land surface is comprised of infertile soils, mainly lateritic clays, lateritic sediments and rockies [106], which has led to it being called Great Red Island. Lateritic soils are the result of the weathering of very diverse rocks (metamorphic, igneous and sedimentary) and most eastern soils are ferralsols [107]. The geomorphology of Madagascar roughly consists of three parallel longitudinal zones: 1) the low highland and plains in the west, 2) the central highland, and 3) the humid coastal strip in the east, where *Hymenaea verrucosa* lives today, mainly on salty–sandy soils (regosols *s.l.*). The thickness of regosols differs depending on the area, however, *Hymenaea* roots at a shallow depth (1 m maximum; Fig 4B and 4C) and even large trees are uprooted during the hurricane season.

According to the soil classification of Oldeman [108] and Jones et al.[109] soils herein studied in the eastern areas belong to the dystric regosols type (Rd, in FAO Soil Units; RGdy *sensu* Jones et al. [109]) and those in the northern area, to the rhodic ferralsols type. In the ancient French classification of the ORSTOM, these soils were described as podzols or pseudo-podzols [110,111]. In the eastern area of study, rainfall exceeds a mean of 1,500 mm a year, temperature averages 24–28˚C, and the growing period lasts more than 270 days. These characteristics are unusual for the formation of regosols which they develop better in arid and semi-arid areas and in mountain regions.

During 2017, two pits were excavated in the Sambava region (Fig 4), both in primary but partially degraded *Hymenaea* forest; one in Andranotsara (Fig 4A and 4B) at 101 m from the coast and 5 m above sea level (Andranotsara pit Q1); the other in Antampolo (Fig 4C) at 1.15 km from the coast and 18 m above sea level (Antampolo pit Q2). Both were excavated in sandy soils (Ø > 0.063 mm) with slight signs of soil development. They showed very poor cohesion and structure coupled with low water retention capacity and scarce levels rich in organic matter. Based on historical documents, together with the observation that *H. verrucosa* roots are superficial yet of great extension, we hypothesised that the rooted soil quickly loses its organic matter.

The soil of Andranotsara pit Q1 (Fig 4A and 4B) developed in ancient coastal sand dunes, resulting in a low level of moisture and nutrients. Digging of the pit Q1 was delimited by three ancient *Hymenaea* tree stumps (trunk diameters of 82 cm, 91 cm and 94 cm, respectively). The trees were affected by two hurricanes, one in 2000 and one in March 2017, and subsequently cut down, leaving their root systems. The soil was mainly sandy and uncemented with a high degree of drainage. Excavation-depth reached 175 cm (Fig 5A). O horizon: 2–3 cm with

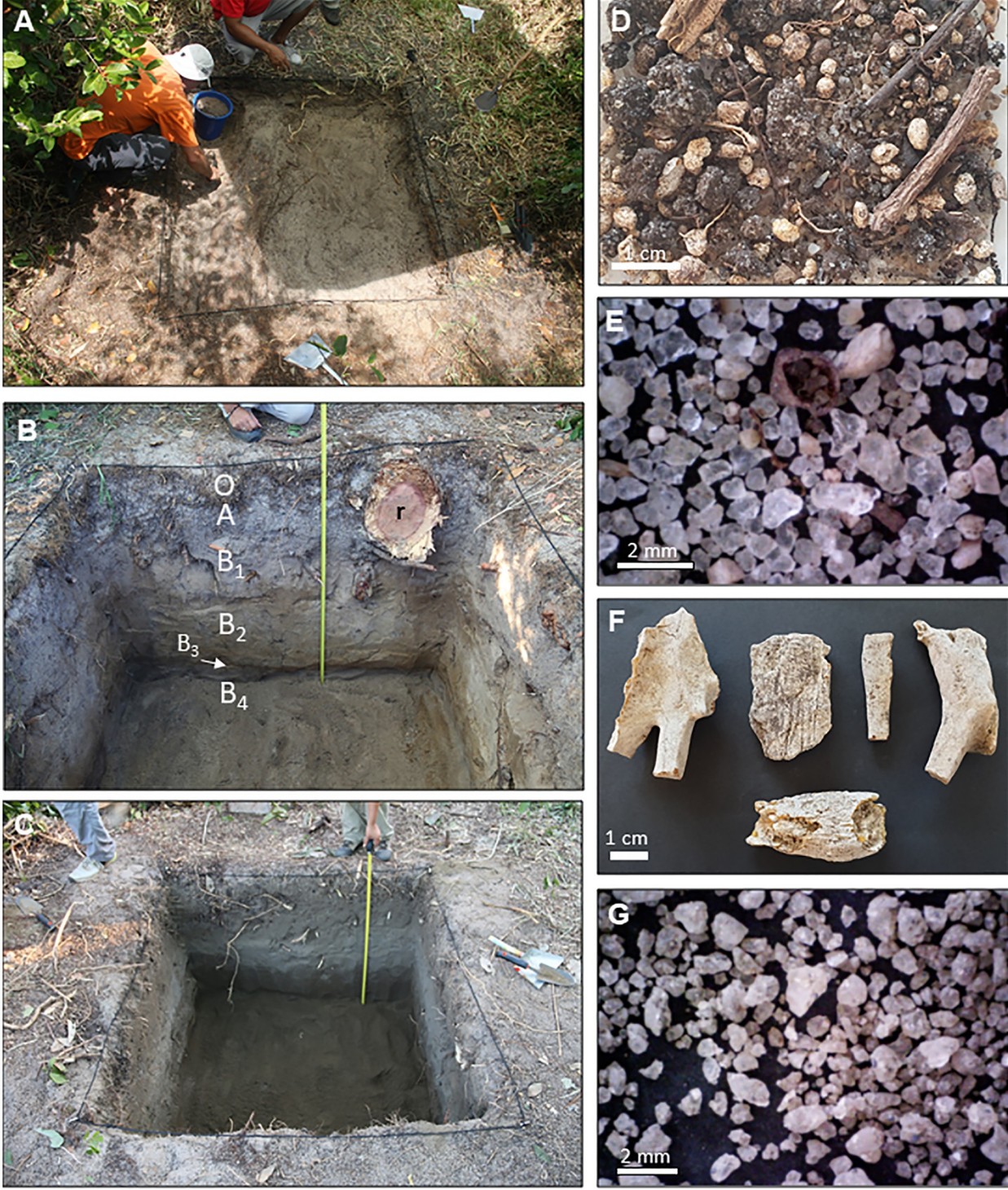

**Fig 4. Pits excavated in the Sambava region.** (A) Commencement of the Andranotsara pit Q1 (S14˚38.472'—E050˚12.467'). (B) Andranotsara pit Q1, showing the different horizons to the top of the sub-horizon $B_4$ (105 cm deep). (C) Final result of the Antampolo pit Q2 (S14˚43.468'—E050˚12.860'), 140 cm deep. (D) Sample from the A horizon of the Andranotsara pit Q1; it includes fungi-bearing resin beads, humus-quartz aggregates, some roots, quartz grains and pellets. (E) Sample from the sub-horizon $B_1$ of the same pit Q1, formed mainly of non-rounded quartz grains, and a small number of fungi-bearing resin beads. (F) Resin remains formed under aerial conditions, found in the Antampolo pit Q2 in the A horizon, showing the intense pulverulent surface alteration. (G) Sample from the sub-horizon $B_3$ of the Antampolo pit Q2, formed of non-rounded quartz grains with clay. Notations: r = *H. verrucosa* root; O, A, $B_1$ to $B_4$ = horizons and sub-horizons.

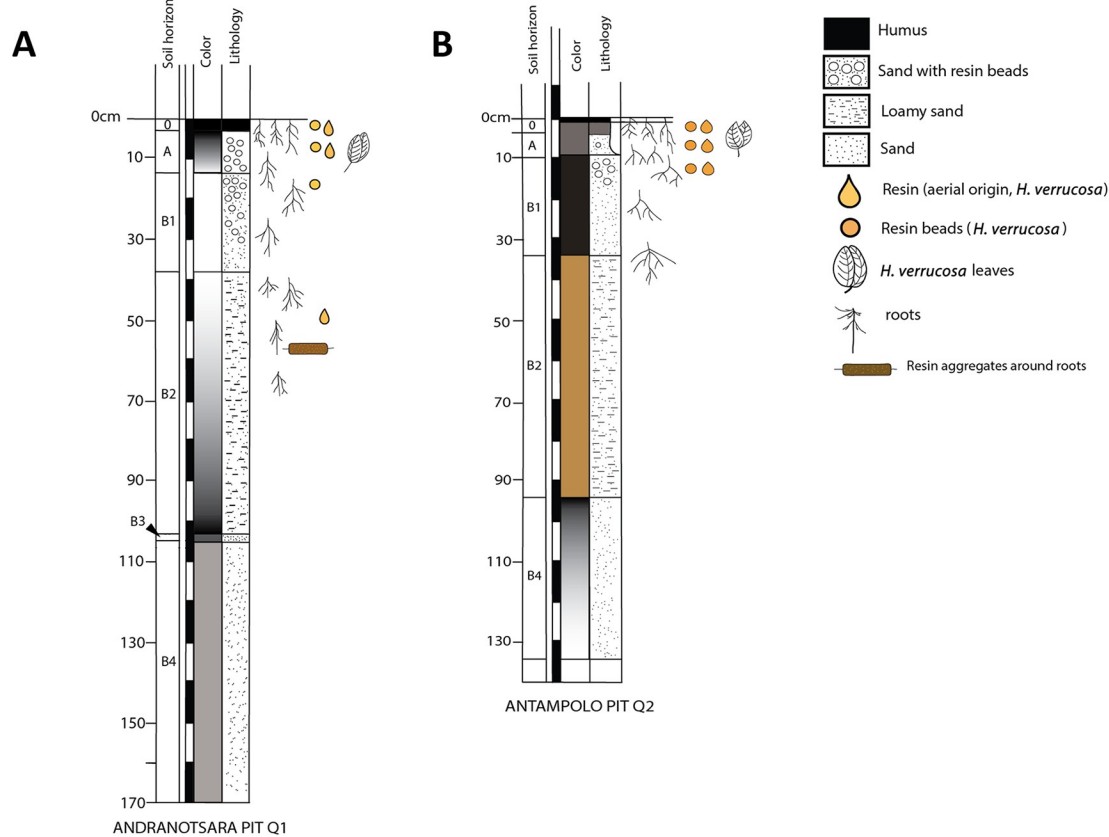

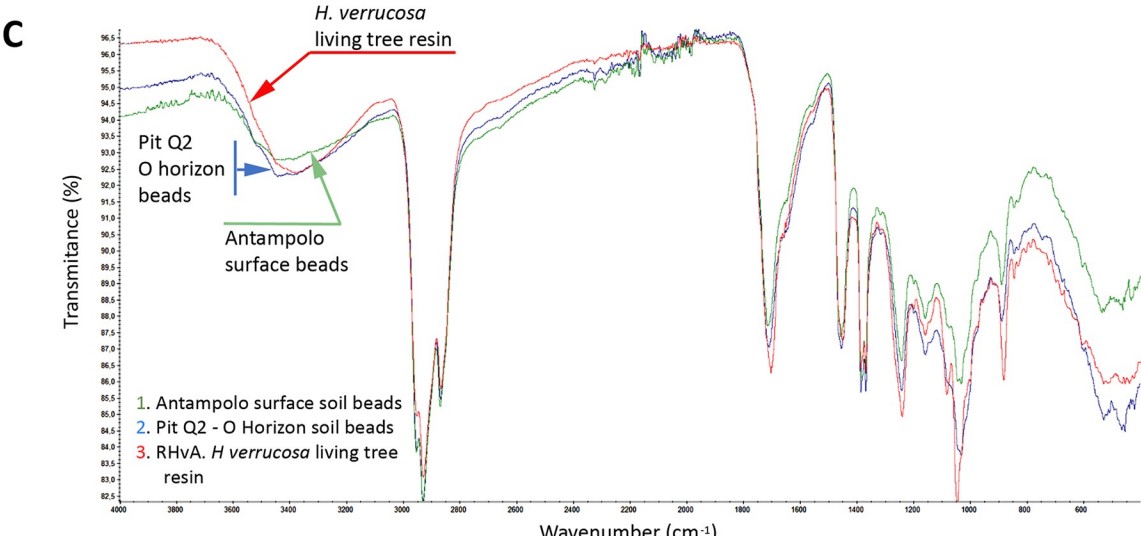

**Fig 5. Pedologic logs of the pits excavated in the Sambava region.** (A) Andranotsara pit Q1, 170 cm deep, and (B) Antampolo pit Q2, 140 cm deep. The detailed explanation is in the text. For geographical locations of the pits, see Fig 4. (C) FTIR analyses of three samples to determine the origin of soil beads (= fungi-bearing resin beads) preserved in the O horizon of the forest soil. The analyses include samples of resin from: 1. soil beads from the soil surface of a different part of Antampolo; 2. soil beads from the O horizon of the Antampolo pit Q2, and 3. living *H. verrucosa* tree resin in Antampolo.

humus on the ground surface, grass roots and fungi-bearing resin beads (referred to above as "soil beads"; see the results in this section below), and fragments of resin pieces formed under aerial conditions (on branches or the trunk). A horizon: the top soil was very dark and rich in organic matter, with a thickness of 10–15 cm, featuring many grass roots, little fragments of *Hymenaea* leaves, abundant fungi-bearing resin beads (Ø 1.9–5.5 mm) (Fig 4D), and some fragments of resin (formed under aerial conditions, < 3 mm). The inorganic fraction is composed mainly of sand rich in quartz. B horizon: the subsoil was divided in four sub-horizons. The sub-horizon $B_1$, 25–30 cm thickness, was composed of transparent quartz sand with some organic matter. Its upper part also contains some fungi-bearing resin beads (Ø <1.1 mm, Fig 4E) and isolated pieces of aerial resin observed also in the lower part of A horizon. Sub-horizon $B_1$ is a level with many *Hymenaea* roots, some with resin wraps that agglutinate the sand. Sub-horizon $B_2$ is a 60–70 cm interval of ochre sand, very monotonous, with unrounded grains, having some roots but with very little organic matter and resin (Fig 4B). Resin and sand aggregates are present around the roots at a depth of 60 cm (Fig 7A and 7B). At this level, the clay content is high, but does not form aggregates. A few grains of igneous rock of a greenish white colour are also present. At 50 cm deep, the last pieces (very few) of preserved resin are located; the ochre tone of $B_2$ intensifies closer to sub-horizon $B_3$, which is a 2 cm dark ochre sandy level with some greenish white grains of igneous rock; from 2 cm to 5 cm there is a change in the colour of the sand to lighter ochre and later to grey with dark spots. Sub-horizon $B_4$, with a minimum thickness of 57 cm, is composed of very clear monotonous sand and apparently lacks roots.

The Antampolo pit Q2 (Fig 4C) is on the side of a hill, developed above a ferralitic soil on top of a plutonic rock (observed laterally, not in the area where the pit was developed). The pit Q2 was dug around the stumps of two *Hymenaea* (trunk diameters of 80 cm and 100 cm, respectively) that were uprooted and cut in 2000, after a hurricane. Excavation depth reached 135 cm (Fig 5B). O horizon: 3 cm of quartz-sand with a high proportion of humus and grass roots, abundant fungi-bearing resin beads and aerial pieces of degraded resin. Neither pod resin beads nor isolated seeds of *H. verrucosa* are observed, perhaps due to rapid decomposition. A horizon: very dark level, rich in organic matter, with a thickness of 6 cm, featuring abundant roots millimetric in section, fungi-bearing resin beads (Ø 1.2–5.4 mm), and some big pieces of resin (some of them developed under aerial conditions, > 5.2 mm) (Fig 4F). B horizon: the subsoil is divided into four sub-horizons. Sub-horizon $B_1$, with a thickness of 24–26 cm, is composed of fine, dark grey sand with some organic matter and small roots. Its upper part also contains some fungi-bearing resin beads (Ø < 1.5 mm) and big fragments of highly altered resin (some related to aerial exudations), not present in the lower part. At the interval between 20 and 30 cm depth, resin crusts occur associated with roots (some of them highly degraded, others with vitreous interior, orange in colour). Sub-horizon $B_2$ is composed of ochre sand that is very monotonous and non-agglutinating clays (Fig 4G), with a thickness of 60 cm. There is no presence of resin in any of its forms. Sub-horizon $B_3$ is composed exclusively of fine sand quartz without clay, with a minimum thickness of 40 cm. C horizon was not reached.

Despite an intensive search in the north, in the Antsiranana region, no copal or resin deposits were found in the margins of the rivers (point bars, oxbows), beaches or deltas involved in the drainage of the *Montagne d'Ambre* (Fig 6A–6D). We consider that this region lacks copal outcrops. It seems more probable that publications mentioning *Montagne d'Ambre* as the source of "Madagascar copal" [16,86,112] followed the unconfirmed information provided by native traders, uncritically, possibly due to spurious interests, because "Madagascar copal" is neither produced nor marketed in the region. *Hymenaea verrucosa* does not grow in the *Montagne d'Ambre* region because of its evergreen tropical forest microclimate. In the drier coastal

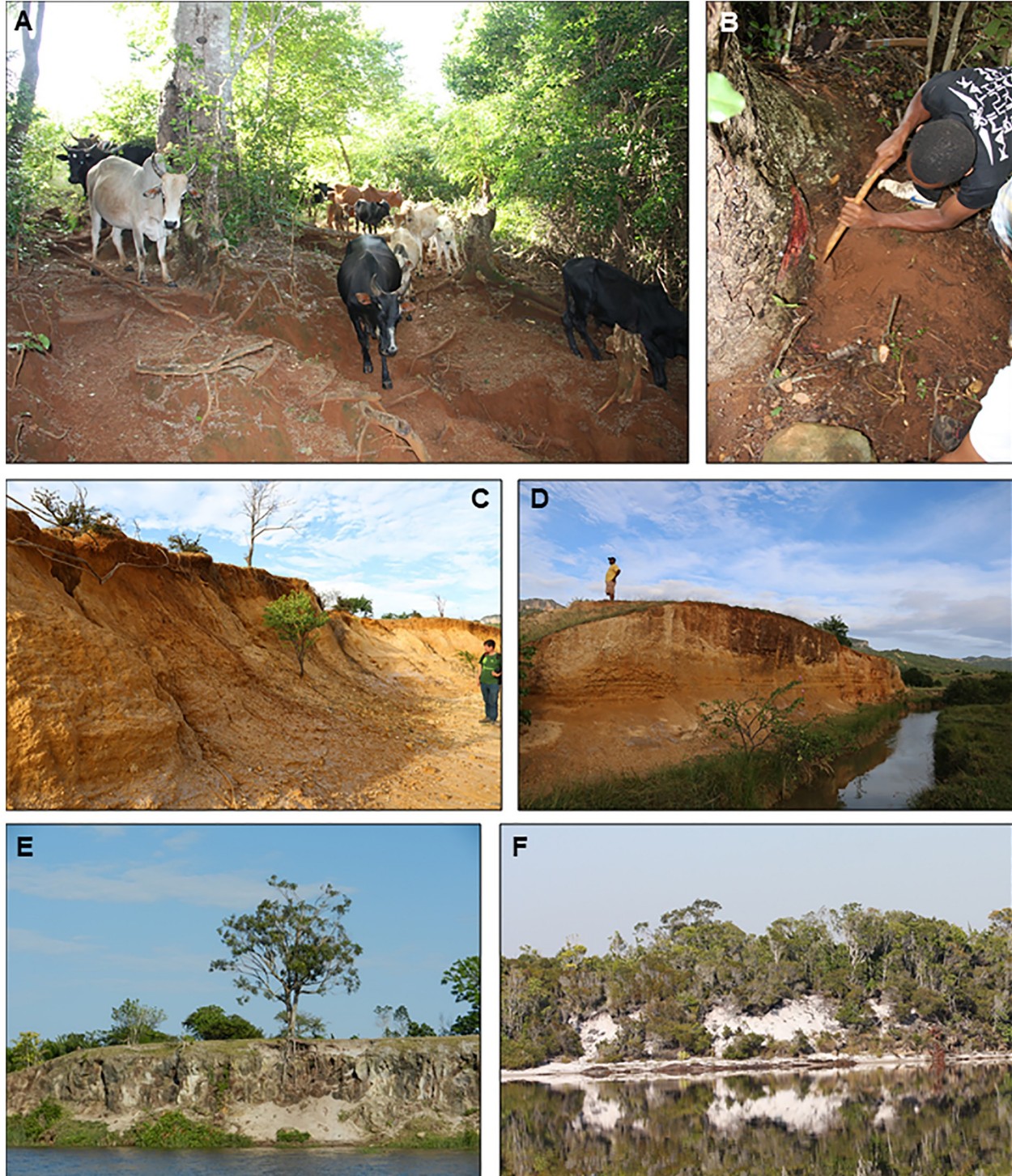

**Fig 6. Soils in which *H. verrucosa* grows.** (A) Ferralsols in Sacaramy (between *Montagne d'Ambre* and Antsiranana). (B) Searching for resin in the litter around an *H. verrucosa* trunk, rooted in a ferralsol, in Sacaramy. (C) Ferralsols in the delta of the river Antomboka, in Anamakia, next to the salt production site of Antsiranana. (D) Ferralsols in a delta plain in front of Nosy Lonjo (Andovobazaha Bay, Antsiranana). (E) Regosols/arenosols in the Pangalanes Channel margins, Ambahy (Nosy Varika, Mananjary). (F) Arenosols in a lake margin between Nosy Varika and Ambahy.

region of Antsiranana—Sakaramy, *H. verrucosa* is nowadays represented only by scattered individuals, which are not forming forests. The edaphic profiles on which *H. verrucosa* lives are ferralitic soils on basalts (rhodic ferralsols) (*sensu* Jones et al. [109]). Here, pits in soil were not dug, but profiles were observed on the margins of the rivers and in some parts of the delta plains of the rivers that delimit Antsiranana (Fig 6C and 6D) draining the *Montagne d'Ambre* area (running through the Sakaramy area to the sea). O horizon: 2–3 cm with humus on ground surface with grass roots without evidence of resin particles. A horizon: a brown clay-sandy level poor in humus, but with numerous roots and with a thickness around 10 cm; neither fungi-bearing resin beads nor pieces of resin developed under aerial conditions have been observed. The first metre of the B horizon is composed of low-consolidated clays which are coarse-textured, and of a brown to red wine colour; the next one and a half metres is a coarse-textured brown–red claystone more heavily consolidated than that above; in the first few centimetres of this horizon the volume of organic matter does not exceed 2%. C horizon: light brown, weathered rock starting abruptly with some reddish spots in places. Spheroidal weathering (small balls), due to the chemical alteration along intersecting joints, is observed. Neither resin nor copal are observed in any ferralitic soil studied. In western Antsiranana we observed some chromic luvisols and in the eastern part, lithic leptosols, both also lacking resin accumulations. In this area we collected resin directly from the trees; *Hymenaea* is not abundant in the Antsiranana region, it has a mystic significance for the local people, and, at least today, trees are dispersed in the region but are not forming forests. Commercial exploitation in resin and copal is unknown in this region.

In Mananhary (Nosy Varika—Ambahy) area, the edaphic profiles are similar to those in Sambava -Antalaha (Andranotsara—Antampolo). Here no pits were dug, but soils were observed in the margins of the Pangalanes Channel (Fig 6E and 6F) and lakes. No resin accumulation levels were observed, only a few pieces of resin in the O and A horizons of dystric regosols/arenosols. Commercial exploitation of *Hymenaea verrucosa* resin is common in this region.

*Hymenaea* roots in sandy soils in areas with a high rainfall rate, where the soils are intensely drained. This drainage means that in a relatively short time the organic matter of the soil, including resin inputs, disappears, mainly through decomposition and degradation [113]. Our investigations into soils in different areas of Madagascar with the presence of *Hymenaea* trees indicate that the high rate of rain combined with a high degree of drainage strongly promotes the decomposition of resin at layers deeper than 50 cm. Resin cannot be preserved in highly washed lateritic soils because these are characterised by a scarcity of silica and high content of iron, alumina and/or other non-removable minerals, constituting a highly oxidizing medium incompatible with the preservation of labile organic matter as resin. Burton [114] and Sunseri [37] noted that "Zanzibar copal" in Tanzania was dug from red lateritic soils, where *Hymenaea verrucosa* had lived several thousand years ago. However, we did not find copal or resin in Malagasy ferralitic soils. Burton [114] mentioned that, according to Arab traders in Tanzania, the redder the soil, the better the copal was for the market. He described the level at which the "copal" was extracted in Tanzania, corresponding to a dark level of humus and hardened clay that varied in thickness from a few centimetres to 45 centimetres, covered by a thin layer of white sand.

In the Malagasy regions we explored, there was no level of white sand covering the subsurface level, which is rich in organic matter; no level rich in organic matter has been found in subsoil under several decimetres of sand. Levels similarly rich in organic matter to those from which the Tanzanians extracted "copal" are present in Madagascar in the first 15 cm below the surface, forming the most superficial horizons of the Malagasy forests soils. It is in this level that most of the resin samples have been found. This suggests that resin pieces found in other

East African regions were possibly protected and preserved by clay sand levels at a depth of several decimetres, which promote the formation of copal deposits. However, this seems not to be the case in Madagascar.

According to Lewton [115],"fossil copal" ("Zanzibar copal" from Tanzania) sometimes exhibited a crust of oxidised material (known as "goose-skin") when it was removed from the ground. The oxidised material consisted of large to small excrescences caused by weathering [116], nevertheless, the interior remained transparent and clear, and varied from pale yellow to brownish in colour. During the field investigations we did not find this kind of crust on resin pieces from the pits; pieces found in the subsoil outwardly displayed a pulverulent, fine white coat (Fig 7G) most probably produced by the action of resinicolous microorganisms such as calicioid lichens and fungi and/or weathering. Some calicioids are clearly specific to certain angiosperms, gymnosperms or microhabitats [117,118]. The fungal mycelia that develop on *Agathis* resin pieces (Gymnospermae: Araucariaceae) deposited on the forest floors make holes that favour degradation and disappearance in less than five years [119]. It is interesting to note that the Malagasy forest is of angiosperms, while the New Caledonia forest is of gymnosperms, and their soil and litter features differed in some aspects [120]. Despite this and considering that both forests have similar annual precipitations and average temperatures (New Caledonia: 2,250 mm and 23˚C) we would expect a similar degree of degradation and disappearance of resin pieces remaining on their soils and leaf litter horizons.

In the O and A horizons of both pits, abundant millimetric rounded pieces invaded by fungi have been obtained (Figs 4D, 7A and 7B). These pieces are very abundant in the soil of the areas studied and are similarly abundantly present in the surface of the crops in these areas. FTIR analyses agree with the identification of *H. verrucosa* resin (Fig 5C) (fungi-bearing resin beads); the fungi-bearing resin beads are not resin contained in the wall of the fruit pods (Fig 2E), which remained on the soil surface as the fruit pods decomposed, as initially hypothesised. Optical microscopy and SEM showed that the core is a resin-free gap (like a big bubble) containing small spherical (Ø ca. 150 μm), organic structures reddish brown in colour (Fig 7C–7E), which do not match the resin beads of the pods (Fig 2E), since the latter are of solid resin (herein named pod resin beads). The fungi-bearing resin beads have a homogeneous shape and are invaded by Recent fungal mycelia, except for a fine external layer (Fig 7F and 7G), with the internal "bubble" containing the small spherical, organic structures that have been interpreted as sclerotia of the same invasive fungus (Fig 7C–7E). The taxonomic identification of yeast and fungi present in the fungi-bearing resin beads is in progress, by means of 16S rRNA gene sequencing (Next-Generation Sequencing), using a combination of culture-dependent and independent techniques, and will be published elsewhere. Preliminary results indicate that the filamentous fungus is most likely to belong to the genus *Aspergillus*, and that taxonomic determination accords well with the presence of the sclerotia in the beads and their morphology [121]. The structure of the mycelium present in the resin body is similar to some resinicolous fungi in the resin from gymnosperm, both extant [122]; own observations from New Zealand and New Caledonia samples) and fossilised [123]. These fungi-bearing resin beads seem to be small resin emissions from the fine, small roots of *Hymenaea*, as they were observed, demonstrating this association, during the pit excavations. Their presence in soil O and A horizons can be explained by the observed abundance of roots of different sizes in the more superficial horizons. We consider that the preservation and abundance of these resin inputs are due to the tree producing them up to its death at a certain soil depth from its roots. For that reason, the beads are not completely degraded in the soils studied, despite the fact that the presence of dense mycelia invading the resin bodies is most likely to promote the degradation of these resin pieces, rather than the other way around. From a taphonomical point of

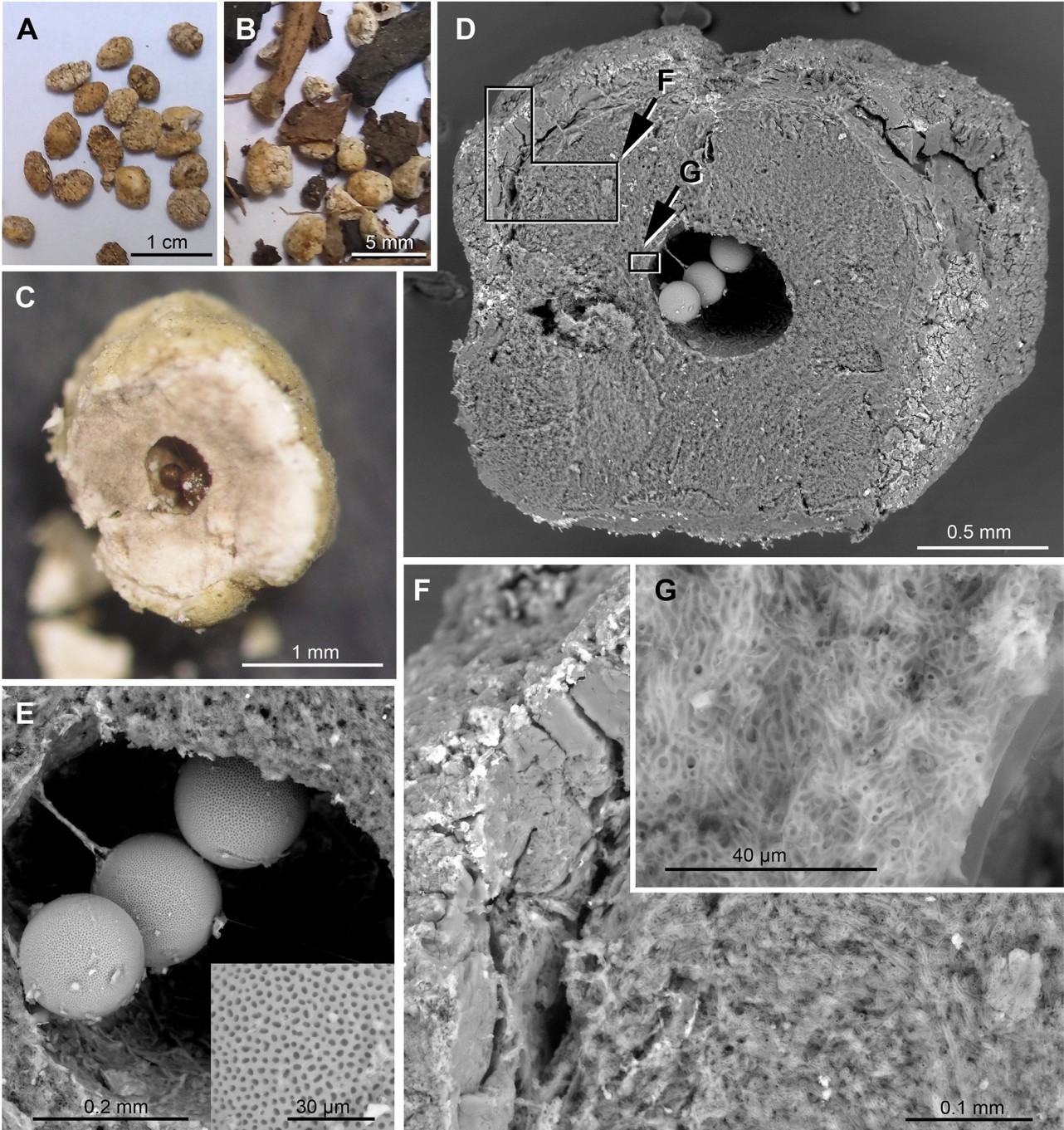

**Fig 7. *Hymenaea verrucosa* fungi-bearing resin beads found abundantly in some pit horizons (featured specimens from pit Q2, Antampolo) are important resin inputs.** (A–B) Fungi-bearing resin beads as organic matter isolated from the sand. (C) Fungi-bearing resin bead, sectioned to show the small spherical, organic structures purple in colour and interpreted as fungal sclerotia. (D) SEM image of the same sectioned bead. (E) SEM detail of the inner cavity containing sclerotia and inset showing a detail of their punctated external surface. (F) SEM detail of the external unaltered layer (left) and the state of preservation in the interior of the resin body of the fungi-bearing bead (right lower part), see inset indicated in D. (G) SEM detail of the resin body of the bead invaded by a fungal mycelium (see inset indicated in D), which is of the same fungus that produced the sclerotia, most probably a representative of the genus *Aspergillus*.

view, these peculiar beads have been observed as important inputs of resin in the *Hymenaea* forest soils of Madagascar.

## The age of "Madagascar copal"

"Madagascar copal" is sold around the world, mainly because it contains preserved arthropods (Fig 8C–8D). Because of its natural clarity and lustre (Fig 8D), merchants often say that it is polished copal, but actually it consists of hardened resin harvested directly from the tree (Fig 8D). Originally, both amber and copal were resin pieces that became buried and underwent fossil diagenetic processes [124]; they are differentiated from their respective original resins by their highest polymerisation grade (see section copal as a scientific term). As explained above, we follow the proposal of Anderson [26], suggesting a scale based on $^{14}C$ dating.

Burleigh and Whalley [125], studied radiocarbon dating of "copal" pieces that came from Tanzania and Kenya (without precise localities), suggesting that they were no older than 100 years. Poinar [13] and Poinar et al. [86] mentioned that "Madagascar copal" (also without precise locality or localities) can be between 50 and 100 years old.

Results reported here are in pMC (percent modern carbon) when the material analysed had more $^{14}C$ than the modern (AD 1950) reference standard, indicating that those samples of "Madagascar copal" were exuded in the last 60 years (pMC) in calibrated calendar years (cal BC/AD) or in Conventional Radiocarbon Age (BP) when the material had less $^{14}C$ than the modern reference standard (Table 2).

Our analyses of $^{14}C$, conducted on these diverse samples from museum pieces and pieces obtained from the pits, suggest that "Madagascar copal" has a relatively small temporal variation in its age (-80 to -300 years), in contrast to previous proposals that postulated ages of between tens and millions of years.

The decomposition of litter can occur rapidly in tropical forests, mainly due to the activity of a great diversity of microbes, fungi and invertebrates, together with high humidity and temperature [126]. In the *Hymenaea* forest, in a sandy soil, with high annual rainfall and, the presence of termites, ants, and other decomposers, organic matter, including resin is preserved for only a short time on the surface [127,128] and is almost absent below a few decimetres depth. We observed, in the pits, that fungi-bearing resin beads (Ø <6 mm) disappeared completely below a depth of a few centimetres. Apparently, all organic matter, including resin, is intensely recycled in a few years, due to the soil washing processes during the seasonally rainy periods and the action of decomposing microorganisms, possibly including resinicolous fungi. During digs in 2017 in Andranotsara, pit Q1, and in Antampolo, pit Q2, aerial resin pieces were very scarce below a depth of 20 cm.

Radiocarbon dating indicates a maximum period of 300 years for the complete degradation of the *Hymenaea* resin fragments in the Malagasy subsoil (Table 2). One piece of resin obtained by us at 50 cm depth in the subsoil has arthropod bioinclusions, however, their degree of preservation (Fig 8E–8G) is notably poorer than in the case of pieces collected directly from the trees (Fig 8C and 8D) and then sold in markets. All pieces collected from the subsoil have a white coat of alteration, as already mentioned, most likely to have been produced in part by fungi and bacteria or by weathering, giving them a matt, pulverulent aspect (Fig 8F). This is probably responsible for resin destruction over time. In the less altered samples, the inner part or core remains vitreous and orange in colour.

The marketed pieces of "Madagascar copal", rich in biota preserved as bioinclusions, came from the Sambava or Mananjary regions, where resin is harvested rather than obtained from the subsoil. Interviews conducted in the different regions revealed that Sambava, Antalaha and Mananjary are the commercial centres for this material, which is collected by people from the

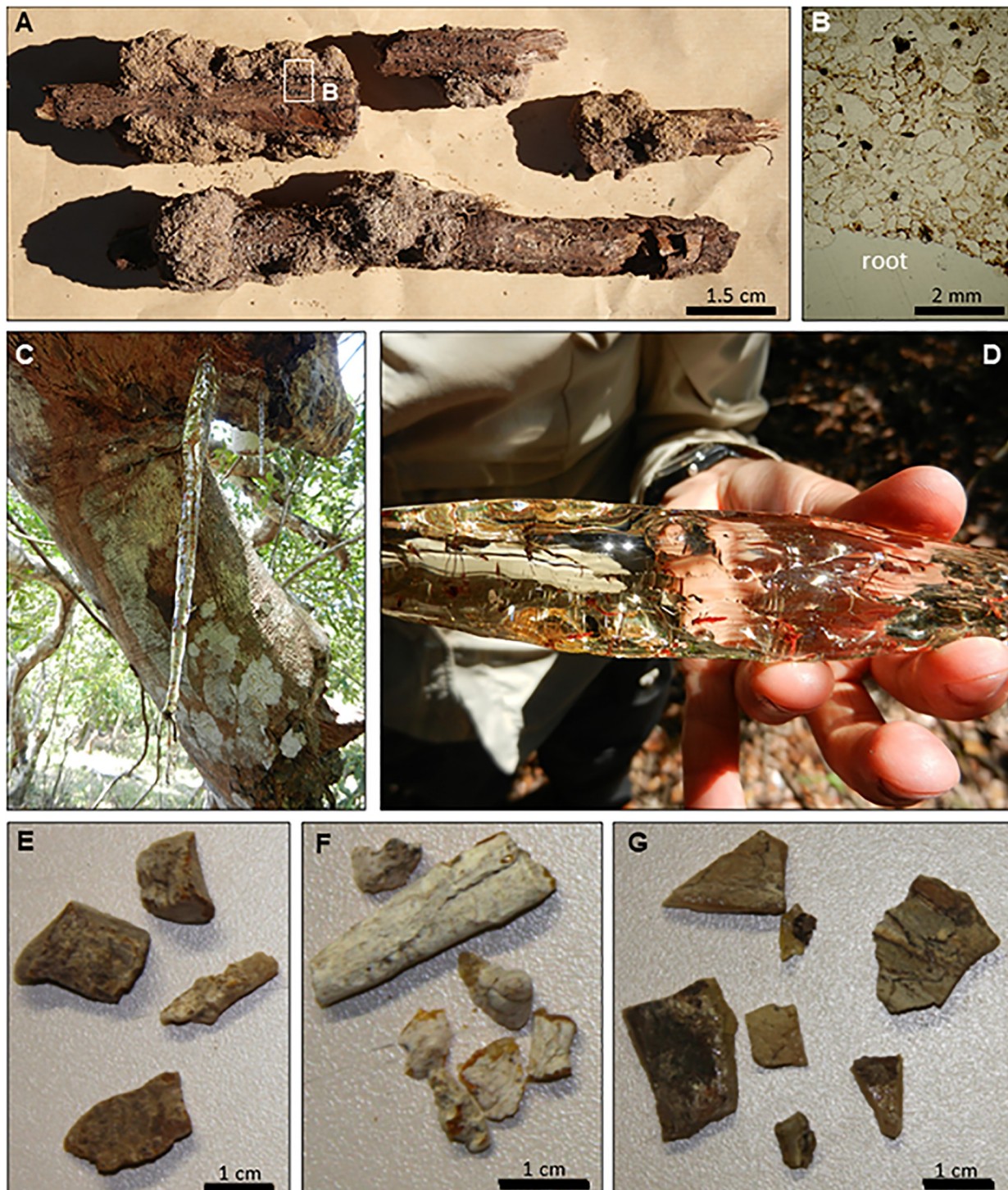

**Fig 8. Pieces of resin obtained from a tree and from the pits at several depths.** (A) Resin exuded by roots around which quartz-grain aggregates form (crusts), as found in the B horizon, Andranotsara pit Q1. (B) Thin section of the A sample; the label "root" shows the position of the *H. verrucosa* root; the resin (brown) when exuded by the root agglutinated the quartz sand grains (transparent). (C) Resin flow of *H. verrucosa* with embedded arthropods, Ambahy (Mananjary). (D) Detail of the same flow showing the original transparency, as if polished. (E) Sample of the resin found in Andranotsara pit Q1 at sub-horizon $B_1$ between 15 cm and 30 cm depth. (F) Sample of the resin found in Antampolo pit Q2 at sub-horizon $B_1$ between 20 cm and 30 cm depth. (G) Sample of the resin found in the same pit at sub-horizon $B_2$ at 50 cm depth. E–G are samples dated by $^{14}$C analysis.

different towns in the surrounding areas and sold to the main traders in an informal market. Resin is collected throughout the year as a supplement to other commercial activities such as vanilla harvest, producing charcoal, etc. The best commercial samples are those collected directly from the trees and not from the soil, because of their clear and lustrous aspect and insect content. This dynamic, domestic trade provides the most important bulk of the resin sold as "Madagascar copal", if not all of it.

Together, these observations and results demonstrate that "Madagascar copal" is of Recent age, with commonly occurring bioinclusions that lead to a commercial misinterpretation with respect to one of its more important putative features. Therefore, it should be considered a modern resin and not an ancient one.

## Palaeobiological implications for previous "Madagascar copal" studies

The first scientific work on arthropods included in "East African copal", registered as "Zanzibar copal", studied a spider [129,130]. Until the 16th century most of the resin or copal varieties, regardless of their origin in East Africa (possibly but not certainly including Madagascar), were denominated in the markets and the scientific works as "Zanzibar copal" (or gum copal), "Calcutta copal", "Bombay copal" or "India copal" [76]. Nevertheless, all these "copals" corresponded to different qualities of the "Zanzibar copal", which was sold in Indian markets [52] had its origin in East Africa and was produced by *H. verrucosa*.

It was not until the end of the 19th century that the first scientific works cited Madagascar as the country of origin of this type of "copal" [131,132]. De Saussure [131] described a bee, endemic to the island, found in "Madagascar copal" *Liotrigona madecassa* (De Saussure, 1890) that today inhabits the western part of the island [133,134]. At the end of the 20th century, due to the worldwide trade in "Madagascar copal", a plethora of arthropods attracted the attention of entomologists and palaeoentomologists, mainly for their country of provenance—a hotspot of biodiversity. This initiated a series of studies lacking crucial data with respect to the geological context(s), the location of the outcrop(s), and the age of the pieces. Some of the species identified from studies of "copal" corresponded to living species that had been previously described [10,80], mainly from the coastal forests of Africa. However, other species are known only from their preservation in "copal" [76,135,136], thus it is possible that some of them could be found alive in the Malagasy eastern coastal forests [10,19]. Recently, Riquelme et al. [137], after a study of scorpions from Miocene Mexican amber, proposed a phylogeny that included three buthid species found in "Madagascar copal"; one species was assigned to the Recent genus *Microcharmus* (Microcharmidae) and the rest to undetermined species in the Recent genus *Palaeogrosphus* [8,136,138]. They suggested that "Malagasy copal" is less than 250 years old (without explanation or new data), and that the species recognised, other than fossil or sub-fossil species, may be considered extant species. The lack of dating of the pieces of "Madagascar copal", including their bioinclusions, is a recurrent but never discussed topic in taxonomic publications. This predicament triggers problems in other research fields such as biogeography [139,140,141], phylogeny [142,143] or molecular clocks [144].

It can be assumed that species that are widespread in Madagascar can be recognised in "copal", however, some taxa may have become extinct because of habitat alteration and/or destruction. Some authors have assumed that "Madagascar copal" has a wide temporal range, suggesting that the oldest samples may include extinct species, while the youngest samples may include species that are still alive [11]. Labandeira [14] considered that "East African copal" (of Tanzanian and Malagasy origin) is a sub-fossil resin 0.7–0.05 Ma old, produced in a subtropical mangrove forest, and including a combination of recently extinct and extant congeneric species, showing a historical structure of modern tropical ecosystems. However, *H.*

*verrucosa* is not part of the mangrove forest in Madagascar or in Tanzania, an ecosystem highly developed on both margins of the Mozambique Channel.

All palaeobiological works must include the age of the deposit where the studied taxa were found. In the case of "Madagascar copal" all the newly described species have been inaccurately located. All specimens studied (between 100 and 120 different species, in about 80 publications) were described under the generic name "Madagascar copal" and several areas of provenance have been proposed. That imprecision is especially problematic for palaeobiological/biological studies, taking into account that *H. verrucosa* inhabits 2,000 km of the east coast of Madagascar with diverse ecosystems (see section material and methods). Thus, specimens can originate from locations up to thousands of kilometres apart. It is also highly probable that mixed specimens from different localities are present in both retail shops and in collections, so putative palaeobiological and/or palaeoenvironmental studies could be based on unsuitable data.

For most of the pieces obtained before the beginning of the 20th century, it is virtually impossible to know if their provenance is in the East African countries such as Tanzania, Mozambique, or Madagascar, because all were sold under the "Zanzibar copal" or "East African copal" names. This absence of accurate geographical information could mean that the studies or inferences in biogeography, ecology, and/or loss/change of biodiversity, among other topics, are inaccurate. This will remain the situation until a technique for recognising the geographical origin of unlabelled or inadequately labelled resin pieces is available.

## Conclusions

*Hymenaea* trees occur in the northern and eastern margins of Madagascar Island. They are most abundant in the regions of Sambava, Antalaha and Masoala, and in the south east at Mananjary (see also Missouri Botanical Garden [145]). Today, resin with arthropod bioinclusions is collected from living, resin producing *Hymenaea verrucosa* trees, by people who sell it in small quantities.

From our studies, we conclude that "Madagascar copal" cannot be considered a sub-fossil resin, it is a Recent resin, which clearly explains the occasional presence of living arthropod species embedded in it. "Madagascar copal" containing remains of preserved organisms and housed in museums and private collections corresponds to Recent resin. Therefore, it does not have the palaeontological relevance that has been attributed to it, however, it is still relevant for actuotaphonomic research. This situation demands a revision of the taxa described to avert both taxonomic errors and inaccurate palaeoenvironmental reconstructions. It is most likely that the species that have been described exclusively from studies of "Madagascar copal" have not been searched for in the appropriate habitat or cannot successfully be caught using conventional entomological traps [6,15]. They can probably be found in the coastal lowland forests where *H. verrucosa* grows, but not among the evergreen tropical forests of the island. However, this observation is based on the premise that the forests have not yet been strongly distorted or even deforested, because the evergreen tropical forest is one of the most endangered habitats in the world.

There is a lack of evidence about how "copal" or resin was extracted in Madagascar during the 18th and 19th centuries; there is only indirect information which comes from ships' captains or military personnel who reported on local trade observed during their trips, or from merchants looking for material for the varnish industry. "Madagascar copal" was known for centuries in the harbours of the Indian Ocean under the trade name "copal of Zanzibar", and later as "Madagascar copal"; however, if it was excavated historically, the extraction localities remain unknown. It was more probably exploited using the same system as today that is,

collected directly from trees or at shallow depths in the ground, and/or from holes excavated to produce charcoal.

The most ancient resin pieces we found in soil are only 300 years old; they were found at < 50 cm depth and considerably altered (resin has not been found deeper). In consequence, the genesis of the Miocene amber deposits in Mexico and the Dominican Republic, in which the *Hymenaea* ssp. resin fossilised, has to have been different from what now exists in the African habitats. Investigations into the formation of amber deposits in Central America, and copal deposits from Colombia and the Dominican Republic could be of great relevance.

It is not ruled out that amber-bearing and/or sub-fossil resin-bearing deposits that originated from ancient exudations of *H. verrucosa* exist in Madagascar. However, the results obtained from the geologically/edaphically studied deposits/soils, the absence of commercially exploited geological deposits, and the current local practice of collecting *H. verrucosa* resin containing arthropods and other biota to sell, suggest their absence. We propose that following this study, any new taxon described from putative "Madagascar copal" must be accompanied by a $^{14}$C analysis of the material in which the bioinclusion is present, at least with respect to the holotype, to establish its validity. For obvious reasons, museums with Malagasy pieces of supposed copal containing type specimens should undertake a review of them by performing $^{14}$C analyses, as was previously suggested [124].

Moreover, even with "Madagascar copal" being a resin produced during the Anthropocene, it could nevertheless contain extinct species due to the high deforestation rate of the island over the last 300 years. Intense current deforestation continues, and perhaps in the next few decades, biota included in Recent *H. verrucosa* resin, together with the historical entomological collections, will be the only reservoirs of knowledge to investigate a part of the entomofauna of the fragile lowland tropical forests of Madagascar.

## Supporting information

**S1 Fig.**
(TIF)

**S2 Fig.**
(TIF)

## Acknowledgments

The authors are grateful to R. Ravelomanana, M. Asensi, S. Rahanitriniaina, T. Rakotondranaivo, M. Madiomanana, and J. Andrianabo for their assistance during the scientific fieldwork, and to J.L. Totovanona and G. Jahovelo for the authorisation to dig in the forests on their property. We thank Dr T. Rakotondrazafy, Director of the Départ. de Paléontologie et Anthropologie Biologique, and Dr E.M. Randrianarisoa, Director of the Départ. d'Entomologie, both in the Université d'Antananarivo, for their help and advice with regards to administrative arrangements. We also thank Dr N. Ferrer from the University of Barcelona's Scientific Facilities for help with the FTIR analysis, R. Kunz (Senckenberg Research Institute and Natural History Museum Frankfurt) for the digitalisation of Figs 1 and 5, Dr T. Hörnschemeyer for providing the C$^{14}$ analysis of one of the analysed pieces, and Alan Lord, both from the same institution for help with the English first text. Furthermore, the authors wish to acknowledge the contribution of the team of the Malagasy Institute for the Conservation of Tropical Environments (ICTE/MICET) who assisted with aspects of the administrative development of our work in Madagascar. We are grateful to Prof Ph. Barden (Editor) and two anonymous reviewers for the comments that have improved the final version of the text.

## Author Contributions

**Conceptualization:** Xavier Delclòs, Enrique Peñalver, Mónica M. Solórzano-Kraemer.

**Data curation:** Xavier Delclòs, Enrique Peñalver, Voajanahary Ranaivosoa, Mónica M. Solórzano-Kraemer.

**Formal analysis:** Xavier Delclòs, Enrique Peñalver, Voajanahary Ranaivosoa, Mónica M. Solórzano-Kraemer.

**Funding acquisition:** Xavier Delclòs, Mónica M. Solórzano-Kraemer.

**Investigation:** Xavier Delclòs, Enrique Peñalver, Mónica M. Solórzano-Kraemer.

**Methodology:** Xavier Delclòs, Enrique Peñalver, Mónica M. Solórzano-Kraemer.

**Project administration:** Xavier Delclòs, Mónica M. Solórzano-Kraemer.

**Resources:** Xavier Delclòs, Voajanahary Ranaivosoa.

**Supervision:** Enrique Peñalver, Mónica M. Solórzano-Kraemer.

**Validation:** Xavier Delclòs, Enrique Peñalver, Mónica M. Solórzano-Kraemer.

**Writing – original draft:** Xavier Delclòs.

**Writing – review & editing:** Enrique Peñalver, Voajanahary Ranaivosoa, Mónica M. Solórzano-Kraemer.

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
