## [Decision Letter · Decision Letter 0]

8 Jan 2020

PONE-D-19-32304

"Madagascar copal": a new approach to study the loss of biodiversity during the Anthropocene

PLOS ONE

Dear Dr Delclos,

Thank you for submitting your manuscript to PLOS ONE. After careful consideration, we feel that it has merit but does not fully meet PLOS ONE’s publication criteria as it currently stands. Therefore, we invite you to submit a revised version of the manuscript that addresses the points raised during the review process.

This is a thorough assessment of Madagascar "copal" that provides excellent new insight into the subject. The authors impressively bring a great deal of data to bear on the subject, including new systematic analyses and a nice summary of history. I am confident that the results will be of interest to a wide community once published. While the reviewers found no fault with any analyses performed or major interpretations, both agreed that the text of the manuscript requires significant improvement. In particular, the authors must work to improve the overall clarity of the text and shorten it where possible. The title must also be reframed to more accurately represent the contents of the paper. I encourage the authors to consider all suggestions made by both reviewers, including those detailed in the attached pdf. In cases where the authors do not wish to include reviewer suggestions, please provide a written response indicating why a change is not necessary. In addition to the reviewer comments, I offer my own minor edits below:

Line 90: The scientific term “copal” is a problem, – ambiguous

Line 141: mainly in the oriental area of – use "Eastern"

Line 167 to 200: in my view, there is value in keeping elements of the history of trade. This recounting may help those with "Madagascar" or "Zanzibar Copal" in collections. However, as both reviewers mention, it would be useful to trim text where available.

Line 713: Remove first "The"

Line 880: Might you mean "oldest" or "ancestral" instead of "most primitive"? The term primitive is imprecise in most contexts.

We would appreciate receiving your revised manuscript by Feb 22 2020 11:59PM. To enhance the reproducibility of your results, we recommend that if applicable you deposit your laboratory protocols in protocols.io, where a protocol can be assigned its own identifier (DOI) such that it can be cited independently in the future. For instructions see: http://journals.plos.org/plosone/s/submission-guidelines#loc-laboratory-protocols

We look forward to receiving your revised manuscript.

Kind regards,

Phillip Barden

Academic Editor

PLOS ONE

Journal Requirements:

2. In your manuscript, please provide additional information regarding the specimens used in your study. Ensure that you have reported specimen numbers and complete repository information, including museum name and geographic location.

For more information on PLOS ONE's requirements for paleontology and archaeology research, see https://journals.plos.org/plosone/s/submission-guidelines#loc-paleontology-and-archaeology-research.

3. We note that Figures 2A and 6B include images of participants in the study. 

"We thank the Spanish Ministry of Economy, Industry and Competitiveness (Project CGL2014-52163), the Spanish Government (Project AEI/FEDER, UE CGL2017-84419), the National Geographic Global Exploration Fund Northern Europe (Project GEFNE 127-14), and the German VolkswagenStiftung (Project N. 90946), for financial support."

6. We note you have included a table to which you do not refer in the text of your manuscript. Please ensure that you refer to Table 1 in your text; if accepted, production will need this reference to link the reader to the Table.

7. Please update your submission to use the PLOS LaTeX template. The template and more information on our requirements for LaTeX submissions can be found at http://journals.plos.org/plosone/s/latex.

Reviewers' comments:

Reviewer's Responses to Questions

**Comments to the Author**

1. Is the manuscript technically sound, and do the data support the conclusions?

Reviewer #1: Yes

Reviewer #2: Yes

2. Has the statistical analysis been performed appropriately and rigorously? 

Reviewer #1: N/A

Reviewer #2: N/A

3. Have the authors made all data underlying the findings in their manuscript fully available?

Reviewer #1: Yes

Reviewer #2: Yes

4. Is the manuscript presented in an intelligible fashion and written in standard English?

Reviewer #1: No

Reviewer #2: Yes

5. Review Comments to the Author

Reviewer #1: This is an important study as it solves the mystery of geographical origin and age of Madagascar copal. The team has much experience in actualistic studies using modern tree resins, and this study is another step forward in the understanding of modern resins and copal. I have, however, several concerns that would need to be addressed prior publication.

The paper is rather extensive, and, for convenience of the readership, all chapters can be shortened without any loss of essential information. The content of some paragraphs could be better be presented as tables to make the data better accessible for the audience. An example is the description of the FTIR data at p. 15, lines 348 – 371.

The title does not reflect the content of the manuscript as the authors do not show a new approach for studying the biodiversity loss of Madagascar during the Anthropocene. A title like ‘Unravelling the mystery of ‘Madagascar copal’: age, origin and preservation of a rather young tree resin’ would much better fit to what is written. It is hard to extract the central theme. The authors should make clear what the actual focus of the study is. Lines 237 – 269 o the palaeobiogeography of Hymenaea, for example, are not at all relevant here. Lines 694 – 710, as currently written, appear as pure speculation.

Many actualistic observations on tree resins in modern forest ecosystems have recently been summarized by Seyfullah et al. 2018 (Biological Reviews). I wonder why this review paper has not been cited.

lines 20-21, ‘In this regard, the study of ancient organisms, like those included in "Madagascar copal", is of relevance.’

Why ancient? The major result of the study is that they are modern.

line 25 should read: However, age and geographical origin …

line 27: replace importance by scientific value

lines 36 and 42: replace hottest hotspot by a meaningful term

line 45: geographically isolated

line 90: replace a problem by problematic

line 333: replace coal by charcoal

line 348: University of Barcelona. Where exactly are the samples housed (department, collection, museum?)

line 379: most likely it came from the area of Mananjary. How do you know? Speculation only?

line 395: replace originated by produced

line 479: The coastal forest harbors a very unusual arthropod community

In what manner is it unique?

line 663: I miss reference to some more recent literature on resinicolous fungi such as Beimforde et al. (2017, Arthropod-Plant Interactions), Rikkinen & Schmidt (2018, in Krings et al. Transformative Palaeontology), and resinicolous bacteria (Saint Martin & Saint Martin 2017, Comptes Rendus Palevol).

line 748: little age variation. 80 to 300 years may appear as large age variation to some readers.

lines 775-778: Did the authors have the opportunity to observe this collection of modern resin from the trees or is this statement based on local hearsay?

line 866: Madagascar is not a habitat! The island contains many habitats, most of them are critically endangered.

The manuscript would greatly benefit from a thorough improvement of the language.

Reviewer #2: This paper provides a clear identification of the geographical origin of the “Madagascar copal”, and demonstrates for the first time clearly the recent age of this resin. The authors combine an intense bibliographic work to a series of analyses and observations. Such work is of major interest given the uncertainties, approximations, and lack of supporting data regarding the age and origin of the “Madagascar copal”. The results lead to reevaluate the work previously done and suggest a new way of investigating the Anthropocene extreme loss of biodiversity in one of the major World “hotspots”.

The paper would benefit from a few modifications. Some comments and suggestions are made in the attached manuscript annotated. A few general comments:

- In my opinion, the title is not reflecting accurately the main focuses of the article defined in the introduction (age and geographical origin of the “Madagascar copal”). Based on the title, I was initially expecting studies of the fauna trapped in the resin, and estimations of the loss of biodiversity in the Anthropocene.

- Unnecessary information. For example: the section Copal as a trade product. I am not convinced about the utility of so much archaeological details for answering your research questions. I would delete most of it, if not all (maybe keep some very basic information and move it to the introduction).

- A few problems of writing style requiring some editing. Some suggestions/corrections are given in the attached manuscript.

- The term “Recent” is frequently used throughout the manuscript to designate non-fossil taxa. I would favor the term “modern” instead.

- The use of quotation marks breaks up the text unnecessarily. It may be better to only use quotation marks around terms like “Madagascar copal” once, at the first mention of the term, and delete all subsequent uses.

6. PLOS authors have the option to publish the peer review history of their article (what does this mean?). If published, this will include your full peer review and any attached files.

Reviewer #1: No

Reviewer #2: No

---

## [Author Response · Author response to Decision Letter 0]

11 Mar 2020

PONE-D-19-32304

"Madagascar copal": a new approach to study the loss of biodiversity during the Anthropocene

Dear Prof Barden,

Next, we answer to the proposed changes, and suggestions for modification by the reviewers to the manuscript (in blue).

Line 90: The scientific term “copal” is a problem, – ambiguous - changed

Line 141: mainly in the oriental area of – use "Eastern" - changed

Line 167 to 200: in my view, there is value in keeping elements of the history of trade. This recounting may help those with "Madagascar" or "Zanzibar Copal" in collections. However, as both reviewers mention, it would be useful to trim text where available. – 

This chapter has been not deleted but its length has been reduced. 

Line 713: Remove first "The" - done

Line 880: Might you mean "oldest" or "ancestral" instead of "most primitive"? The term primitive is imprecise in most contexts. 

Most primitive is not imprecise in this case because we refer to Hymenaeae verrucosa, that is considered in all phylogenetic analyses’ sister of all other 14 species of the genus.

• A rebuttal letter that responds to each point raised by the academic editor and reviewer(s). This letter should be uploaded as separate file and labeled 'Response to Reviewers'. Done

• A marked-up copy of your manuscript that highlights changes made to the original version. This file should be uploaded as separate file and labeled 'Revised Manuscript with Track Changes'. Done

• An unmarked version of your revised paper without tracked changes. This file should be uploaded as separate file and labeled 'Manuscript'. Done

We wish to make the peer review history publicly available to possible readers.

Journal Requirements:

Done.

2. In your manuscript, please provide additional information regarding the specimens used in your study. Ensure that you have reported specimen numbers and complete repository information, including museum name and geographic location. Done.

'All necessary permits were obtained for the described study, which complied with all relevant regulations.' Done, all permits requested and granted are included in the acknowledgments section.

If no permits were required, please include the following statement: 'No permits were required for the described study, which complied with all relevant regulations.'

For more information on PLOS ONE's requirements for paleontology and archaeology research, see https://journals.plos.org/plosone/s/submission-guidelines#loc-paleontology-and-archaeology-research.

3. We note that Figures 2A and 6B include images of participants in the study. 

We preferred to change the images using other in which no face is observed, because we cannot contact the people in the Figures 2A and 6B. 

Figures changed.

"We thank the Spanish Ministry of Economy, Industry and Competitiveness (Project CGL2014-52163), the Spanish Government (Project AEI/FEDER, UE CGL2017-84419), the National Geographic Global Exploration Fund Northern Europe (Project GEFNE 127-14), and the German VolkswagenStiftung (Project N. 90946), for financial support."

Done. Funding information has been removed from the text. It is now included in the Funding Statement section on the online submission form.

Xavier Delclòs: I have a serious problem validating my ORCID code in the PlosOne platform. When I try to authenticate the ORCID identifier through the Fetch / Register link, the PLOS application does not contact the ORCID application. My ORCID code is: 0000-0002-2233-5480 and I agree if you include it in the paper.

6. We note you have included a table to which you do not refer in the text of your manuscript. Please ensure that you refer to Table 1 in your text; if accepted, production will need this reference to link the reader to the Table.

The two tables included in the manuscript are now cited in it.

Reviewers' comments:

Reviewer's Responses to Questions

Comments to the Author

1. Is the manuscript technically sound, and do the data support the conclusions?

Reviewer #1: Yes

Reviewer #2: Yes

2. Has the statistical analysis been performed appropriately and rigorously?

Reviewer #1: N/A

Reviewer #2: N/A

 3. Have the authors made all data underlying the findings in their manuscript fully available?

Reviewer #1: Yes

Reviewer #2: Yes ________________________________________

 4. Is the manuscript presented in an intelligible fashion and written in standard English?

Reviewer #1: No

Reviewer #2: Yes

We thank the two reviewers and the Editor for the effort in improving the English text. The current revised manuscript has been reviewed by the translation service of the University of Barcelona to improve the English. 

5. Review Comments to the Author

Reviewer #1: 

1) The paper is rather extensive, and, for convenience of the readership, all chapters can be shortened without any loss of essential information. The content of some paragraphs could be better be presented as tables to make the data better accessible for the audience. An example is the description of the FTIR data at p. 15, lines 348 – 371. 

Some chapters have been shortened and now FTIR data are presented as a Table (Table 1)

2) The title does not reflect the content of the manuscript as the authors do not show a new approach for studying the biodiversity loss of Madagascar during the Anthropocene. A title like ‘Unravelling the mystery of ‘Madagascar copal’: age, origin and preservation of a rather young tree resin’ would much better fit to what is written. It is hard to extract the central theme. The authors should make clear what the actual focus of the study is. 

The title of the manuscript has been changed as suggested this reviewer. 

3) Lines 237 – 269 o the palaeobiogeography of Hymenaea, for example, are not at all relevant here. 

We consider of high relevance the indication in this manuscript that the genus Hymenaea in Madagascar is present since the Miocene, and not latter, and that the lack of amber deposits from Hymenaea in Madagascar contrast to the rich Miocene amber deposits from Mexico or Dominican Republic. This circumstance only can be explained as different geological and environmental conditions. We preferred to maintain the original text.

4) Lines 694 – 710, as currently written, appear as pure speculation.

The speculative sentences about the possible origin of the beads in fungal infection and possible relationship of the fungus and the tree have been deleted. The other part of this section has been changed and now there is indication that the statements it contains are based on our taphonomic observations during the pit excavations and on the soil surfaces around.

5) Many actualistic observations on tree resins in modern forest ecosystems have recently been summarized by Seyfullah et al. 2018 (Biological Reviews). I wonder why this review paper has not been cited.

Our work analysed specifically the age of the copal from Madagascar. Seyfullah et al. (2018) is a very nice summary about the state-of-the-art of investigations about resin production, amber deposits, and amber and resin classification done during the last years but in our opinion, it is not relevant for the content of our manuscript. 

6) lines 20-21, ‘In this regard, the study of ancient organisms, like those included in "Madagascar copal", is of relevance.’ Why ancient? The major result of the study is that they are modern. 

We deleted the word “ancient”. In this context, the age of the resin is not relevant. Is important to mention that the study of the inclusions in Madagascar resin is important because “Madagascar copal” can be about 300 years old. In a region with a very high rate of biodiversity loss, such as the lowland coastal forest of the eastern Madagascar, due to intense anthropic activity, it is possible that some species present in resin have already disappeared.

7) line 25 should read: However, age and geographical origin … - replaced

8) line 27: replace importance by scientific value - replaced

9) lines 36 and 42: replace hottest hotspot by a meaningful term 

We consider it do not to proceed. The term “hottest hotspots” was defined by Myers et al. (2000) in Nature for the eight unusual rich hotspots on biodiversity in the world, and Madagascar is one of the so-called “hottest hotspot”. 

Myers et at. (2000). Biodiversity hotspots for conservation priorities. Nature 403: 853–858.

10) line 45: geographically isolated – we are included the term “geographically” 

11) line 90: replace a problem by problematic – we are replaced “problem” by “ambiguous” as the Editor suggested.

12) line 333: replace coal by charcoal - replaced

13) line 348: University of Barcelona. Where exactly are the samples housed (department, collection, museum?). 

We are included in the text a more precise location of the samples.

14) line 379: most likely it came from the area of Mananjary. How do you know? Speculation only? 

Yes, it is not sure the precedence of these pieces. However, in the Mananjary region there is a large current resin trade. Resin is collected in this area for sale as "Madagascar copal" and it is relatively close to the city of Antsirabe (almost 150 km away), at least, much closer than the Sambava region that are almost 1000 km away. It is for this reason that we assume that the material studied comes from Mananjary, and not from the north of Madagascar, the other area where resin is collected for sale.

15) line 395: replace originated by produced - replaced

16) line 479: The coastal forest harbors a very unusual arthropod community - In what manner is it unique? 

Madagascar's east lowland forest degrades at high speed because it is one of the most populated areas, but at the same time it is one of the least studied since research teams focus their studies on the rainforest, where there is a greater diversity of flora and fauna. The fauna of arthropods that live in this area is quite different from what is usual to visualize in Malagasy entomological treatises.

17) line 663: I miss reference to some more recent literature on resinicolous fungi such as Beimforde et al. (2017, Arthropod-Plant Interactions), Rikkinen & Schmidt (2018, in Krings et al. Transformative Palaeontology), and resinicolous bacteria (Saint Martin & Saint Martin 2017, Comptes Rendus Palevol). 

Any of these references do not refer or have relation to the copal of Madagascar. For this reason, we prefer not included them in the manuscript.

18) line 748: little age variation. 80 to 300 years may appear as large age variation to some readers. 

We modified the sentence in order to precise the age: It has a relatively small temporal variation in its age (-80 to -300 years), in relation to the previous proposals that postulate between tens and millions of years. 

19) lines 775-778: Did the authors have the opportunity to observe this collection of modern resin from the trees or is this statement based on local hearsay? 

Resin harvesting is common in the regions of Mananjary (Nosy Varika, 2013) or Sambava (Andranotsara, 2017) where the people who collect resin from Hymenaea verrucosa came to offer us - pieces containing insects. It is true that we had a Malagasy guide who encouraged to the locals. Local people offered resin amounts containing stalactite-shaped pieces over 50 cm long plenty of insect inclusions; however, our affirmation is not based on local hearsay, but direct observation in the field. We also noticed in the field that the material present in the forest soils is poorly preserved. The pieces sold in the markets have not been polished, as we were informed, but they were collected directly from the trees. Hymenaea resin polymerizes within a few days and remains transparent until its burial. We reiterate that pieces sold as polished "copal" are fresh resin directly collected from the trees without preparation.

19) line 866: Madagascar is not a habitat! The island contains many habitats, most of them are critically endangered. 

The word "in" has been erased. The sentence refers to the lowland coastal forest area on the Eastern Madagascar, as one of the most degraded habitats of the island.

20) The manuscript would greatly benefit from a thorough improvement of the language. 

The manuscript was reviewed by an English researcher before first submission to the journal. However, before resubmission the new version to PlosOne was corrected by the Language Services of the University of Barcelona, in order to improve the English.

Reviewer #2: 

The paper would benefit from a few modifications. Some comments and suggestions are made in the attached manuscript annotated. A few general comments:

1) In my opinion, the title is not reflecting accurately the main focuses of the article defined in the introduction (age and geographical origin of the “Madagascar copal”). Based on the title, I was initially expecting studies of the fauna trapped in the resin, and estimations of the loss of biodiversity in the Anthropocene. 

We replaced the title exactly as suggested the reviewer #1.

2) Unnecessary information. For example: the section Copal as a trade product. I am not convinced about the utility of so much archaeological details for answering your research questions. I would delete most of it, if not all (maybe keep some very basic information and move it to the introduction). 

We consider the historical information we provide in order to differentiate the moment when the trade of the "Madagascar copal" is started to be individualized with respect to the "Zanzibar copal" or "East African copal" it is essential to know from what date we can carefully differentiate the origin of the historical material housed in museums. However, this part has been summarized leaving what we considered more essential.

A few problems of writing style requiring some editing. Some suggestions/corrections are given in the attached manuscript.

All these changes have been accepted. Also, some sentences have been rewriting, in order to improve understanding.

3) The term “Recent” is frequently used throughout the manuscript to designate non-fossil taxa. I would favor the term “modern” instead. – 

The use of the terms “Recent” or “modern” in the text have been reviewed by the correction service of the University of Barcelona, in order to be used each word in its place properly.

4) The use of quotation marks breaks up the text unnecessarily. It may be better to only use quotation marks around terms like “Madagascar copal” once, at the first mention of the term, and delete all subsequent uses. 

If it is not required by the Journal or the Editor, we consider that quotation marks must be kept because the name “Madagascar copal” is mention as a literal name through the whole manuscript. Please, note that the main conclusion of this manuscript is that it is not copal, thus we prefer to maintain that notation depending of the progress of the manuscript content and the contexts in each case. In respect to this, we included these quotation marks very carefully in the original manuscript.

---

## [Decision Letter · Decision Letter 1]

7 Apr 2020

PONE-D-19-32304R1

Unravelling the mystery of “Madagascar copal’: age, origin and preservation of a Recent resin

PLOS ONE

Dear Dr Delclos,

Thank you for submitting your manuscript to PLOS ONE. After careful consideration, we feel that it has merit but does not fully meet PLOS ONE’s publication criteria as it currently stands. Therefore, we invite you to submit a revised version of the manuscript that addresses the points raised during the review process.

Both reviewers agree that the manuscript is much improved and should be published. However, both identified small errors or omissions that should be amended prior to final acceptance. I am issuing a decision of "minor revision" as a mechanism for incorporating minor reviewer comments, however the manuscript will not have to undergo an additional round of reviews. Once the revised version is resubmitted, so long as the reviewer suggestions are incorporated or responded to appropriately, the manuscript will receive a final decision of "accept."

Reviewer comments relate to typos or important references that should to be included. With respect to suggested citations, unless the authors are very strongly opposed for reasons related to poor fit with the current manuscript, I suggest including them. This publication will likely serve as a reference for some time; having thorough and up-to-date references will only improve its usefulness for other researchers. 

Additionally, I have one comment as a follow-up to the authors' response: 

Line 843: the term primitive here is in fact imprecise in a phylogenetic context. I recommend stating "species sister to all remaining Hymenaea species". Please see this publication for a discussion on "primitive" in phylogenetics: Omland, K.E., Cook, L.G. and Crisp, M.D., 2008. Tree thinking for all biology: the problem with reading phylogenies as ladders of progress. BioEssays, 30(9), pp.854-867.

We would appreciate receiving your revised manuscript by May 22 2020 11:59PM. To enhance the reproducibility of your results, we recommend that if applicable you deposit your laboratory protocols in protocols.io, where a protocol can be assigned its own identifier (DOI) such that it can be cited independently in the future. For instructions see: http://journals.plos.org/plosone/s/submission-guidelines#loc-laboratory-protocols

We look forward to receiving your revised manuscript.

Kind regards,

Phillip Barden

Academic Editor

PLOS ONE

Reviewers' comments:

Reviewer's Responses to Questions

**Comments to the Author**

1. If the authors have adequately addressed your comments raised in a previous round of review and you feel that this manuscript is now acceptable for publication, you may indicate that here to bypass the “Comments to the Author” section, enter your conflict of interest statement in the “Confidential to Editor” section, and submit your "Accept" recommendation.

Reviewer #1: (No Response)

Reviewer #2: All comments have been addressed

2. Is the manuscript technically sound, and do the data support the conclusions?

Reviewer #1: Yes

Reviewer #2: Yes

3. Has the statistical analysis been performed appropriately and rigorously? 

Reviewer #1: N/A

Reviewer #2: N/A

4. Have the authors made all data underlying the findings in their manuscript fully available?

Reviewer #1: Yes

Reviewer #2: Yes

5. Is the manuscript presented in an intelligible fashion and written in standard English?

Reviewer #1: Yes

Reviewer #2: Yes

6. Review Comments to the Author

Reviewer #1: I am pleased that the authors considered my title suggestion and at least some of my other concerns.

I recommended citing Seyfullah et al. (2018) as is not exclusively a review paper but as it also reports many new observations. It discusses and illustrates preservation versus degradation of resins in soils of humid tropical forest (including observed time of complete resin degradation), also considering the relevance of resinicolous fungi. Refer, for example, to the ‘Formation of amber deposits’ chapter, tracing the path from resin exudation to the deposit. See also fig. 13. where resin on and inside forest soils is shown from the Pacific region and fig. 8 for resin preserved in peat swamps. The paragraphs on pages 1705-1706 have thus direct relation to the current study. If this paper is not worth citing here, the authors could delete approximately 30% of the presently cited literature from the manuscript.

I recommended citing the literature on resinicolous fungi and bacteria because the authors have randomly chosen the rather old study by Tibell & Titov (1995) which is important but not necessarily relevant for the current study. Speranza et al. (2015) studied supposed fungi from Cretaceous amber from Spain and (like Seyfullah et al. 2018, which is not considered by the authors to have any relation to the current study) from modern resin of New Zealand. Some of the results provided by Speranza et al. (2015) turned out to be controversial and have been revised by Saint Martin & Saint Martin (2017). Thus, the reasoning of the authors for excluding these references is not comprehensible.

I recommended citing Rikkinen & Schmidt (2018) because this is the most recent ecological study on fossil and modern resinicolous fungi with reference to their relevance for interpretation of the forest ecosystems, as calicioid fungi serve as indicator taxa for modern old-growth forests. I recommended citing Beimforde et al. (2017) because it is likewise a direct observation of resinicolous fungi (and their interactions with insects) in a modern tropical forest ecosystem dominated by very resinous plants.

line 446: ‘The coastal forest harbors a very unusual arthropod community’ – My previous question was: In what manner is it unique?

Thanks for explaining it to me, but the authors should likewise explain it to the readers.

Reviewer #2: The comments and suggestions of both reviewers have been taken into account. I feel that the manuscript has been improved and gained significantly in clarity. Therefore, I recommend the acceptance.

Although I haven’t specifically proofread the manuscript for spelling errors, I noticed a few things:

Line 57: write “detail” instead of “detain”

Line 81: “hinders” instead of “hiders”

Line 113: “one-million-year-old” instead of “one-million-years-old”

Line 271: add a space = “between 10,000” instead of “between10,000”

Line 478: “lasts” instead of “last”

Line 533 (figure caption): add a space = “from: 1.” instead of “from:1.”

Line 587 (figure caption): replace “Montage” by “Montagne”

Line 599: italicize the entire species name

Lines 684/685: Incorrect phrasing. Did you mean: ”As explained above, we followed the proposal of Anderson (1996) suggesting a scale based on 14C dating”?

Line 737: replace “through” by “throughout”

Line 743: delete one “that” (“that lead” instead of “that that lead”)

Besides, it doesn’t look like the Table 1 is called in the text. Maybe it should be called at line 706? The sentence “The samples radiocarbon dated are the following:” is not followed by anything. A new paragraph starts after.

7. PLOS authors have the option to publish the peer review history of their article (what does this mean?). If published, this will include your full peer review and any attached files.

Reviewer #1: No

Reviewer #2: No

---

## [Author Response · Author response to Decision Letter 1]

17 Apr 2020

RESPONSE TO REVIEWERS

EDITOR

Additionally, I have one comment as a follow-up to the authors' response: 

Line 843: the term primitive here is in fact imprecise in a phylogenetic context. I recommend stating "species sister to all remaining Hymenaea species". Please see this publication for a discussion on "primitive" in phylogenetics: Omland, K.E., Cook, L.G. and Crisp, M.D., 2008. Tree thinking for all biology: the problem with reading phylogenies as ladders of progress. BioEssays, 30(9), pp.854-867.

Thanks for suggesting this article to us; it is very well written and clarifies some concepts. In general, we work in phylogenetic analysis, so we are familiar with the term’s "sister" and "primitive", but the article really makes us think about how to interpret some phylogenetic hypotheses. Based on some phylogenetic proposals (Langenheim and Lee, 1974 or Fougère-Danezan, 2006) we included in the MS that Hymenaea verrucosa was "the most primitive species of Hymenaea, and from which all the Neotropical species derived". Hymenaea verrucosa was considered primitive compared to the other species of the genus by Langenheim and Lee, who proposed the first phylogenetic hypotheses of the genus, based on the structure of the flowers. It was Fougère-Danezan who proposed H. verrucosa as a sister to all the other species of Hymenaea, with Neotropical distribution. After reading the proposed paper, we consider that perhaps our text was not accurate. 

However, the new proposal by De la Estrella et al. (2018) proposes H. verrucosa as a sister to H. oblongifolia from northern South America and these two as sister to all other species of the genus. This new phylogenetical proposal does not affect our conclusions. Since the MS is not concerned with the Hymenaea phylogeny and evaluating the different proposals, we prefer, if you agree, to delete this sentence in the final version.

• De la Estrella, M., Forest, F., Klitgård, B., Lewis, G.P., Mackinder, B.A., de Queiro, D.P., Wieringa, J.J., Bruneau, A. A new phylogeny-based tribal classification of subfamily Detarioideae, an early branching clade of florally diverse tropical arborescent legumes. Scientific Reports. 2018; 8:6884.

• Fougère-Danezan, M., Herendeen, P.S., Maumont, S., Bruneau, A. Morphological evolution in the variable resin-producing Detarieae (Fabaceae): do morphological characters retain a phylogenetic signal? Annals of Botany. 2010; 105:311–325.

• Langenheim, J.H., Lee, Y.T. Reinstatement of the genus Hymenaea (Leguminosae: Caesalpinioideae) in Africa. Brittonia. 1974; 26:3–21.

Comments to the Author

1. If the authors have adequately addressed your comments raised in a previous round of review and you feel that this manuscript is now acceptable for publication, you may indicate that here to bypass the “Comments to the Author” section, enter your conflict of interest statement in the “Confidential to Editor” section, and submit your "Accept" recommendation.

Reviewer #1: (No Response)

Reviewer #2: All comments have been addressed

2. Is the manuscript technically sound, and do the data support the conclusions?

Reviewer #1: Yes

Reviewer #2: Yes

3. Has the statistical analysis been performed appropriately and rigorously?

Reviewer #1: N/A

Reviewer #2: N/A

4. Have the authors made all data underlying the findings in their manuscript fully available?

Reviewer #1: Yes

Reviewer #2: Yes ________________________________________

5. Is the manuscript presented in an intelligible fashion and written in standard English?

Reviewer #1: Yes

Reviewer #2: Yes

6. Review Comments to the Author

Reviewer #1: 

I am pleased that the authors considered my title suggestion and at least some of my other concerns.

I recommended citing Seyfullah et al. (2018) as is not exclusively a review paper but as it also reports many new observations. It discusses and illustrates preservation versus degradation of resins in soils of humid tropical forest (including observed time of complete resin degradation), also considering the relevance of resinicolous fungi. Refer, for example, to the ‘Formation of amber deposits’ chapter, tracing the path from resin exudation to the deposit. See also fig. 13. where resin on and inside forest soils is shown from the Pacific region and fig. 8 for resin preserved in peat swamps. The paragraphs on pages 1705-1706 have thus direct relation to the current study. If this paper is not worth citing here, the authors could delete approximately 30% of the presently cited literature from the manuscript.

Obviously, we totally disagree with the reviewer's last comment. In the first revision we did not cited some references proposed by this reviewer because all the observations make by their authors are of resins produced by gymnosperms. It is well known that the evolution of soils under angiosperm and gymnosperm forests are different, due to the rapid and slow degradation of their OM, respectively. 

Nevertheless, because all the observations made by Seyfullah et al. (2008) were of the soil floor and leaf-litter, and because the climate are similar in our area and in the area studied by Seyfullah and collaborators, we would expect that the degradation and possible disappearance of the resin bodies are similar. For this reason, we include in the MS a brief new paragraph explaining that and introducing the research by Seyfullah et al. (2008). 

I recommended citing the literature on resinicolous fungi and bacteria because the authors have randomly chosen the rather old study by Tibell & Titov (1995) which is important but not necessarily relevant for the current study. Speranza et al. (2015) studied supposed fungi from Cretaceous amber from Spain and (like Seyfullah et al. 2018, which is not considered by the authors to have any relation to the current study) from modern resin of New Zealand. Some of the results provided by Speranza et al. (2015) turned out to be controversial and have been revised by Saint Martin & Saint Martin (2017). Thus, the reasoning of the authors for excluding these references is not comprehensible.

We introduced the reference of Tibell & Titov (1995) because they explained the presence of resiniferous fungi on angiosperm exudates. It is true that we introduced our publication (Speranza et al. 2015) because some resiniferous Cretaceous fungi are discovered affecting amber pieces of Spain (araucariacean origin), and it is also true that some results turned out to be controversial for some other researchers, but because our research in Madagascar is only on angiosperm resins, we prefer do not cite both references (Speranza et al., 2015 and Saint Martin and Saint Martin, 2017). However, we include the reference Rikkinen & Schmidt (2018) because they explained the presence of calicioid lichens and fungi on the angiosperm exudates.

I recommended citing Rikkinen & Schmidt (2018) because this is the most recent ecological study on fossil and modern resinicolous fungi with reference to their relevance for interpretation of the forest ecosystems, as calicioid fungi serve as indicator taxa for modern old-growth forests. I recommended citing Beimforde et al. (2017) because it is likewise a direct observation of resinicolous fungi (and their interactions with insects) in a modern tropical forest ecosystem dominated by very resinous plants.

We consider that the inclusion of the reference Beimforde et al. (2017) is not suitable in the manuscript because that research is not related with our case study. These authors explained that the resin production of Araucaria humboldtensis is closely related to the borer activity of the larvae of two different group of beetles. One species of ascomycete and diverse fungi growth in the surface of these gymnosperm exudates. The strong host speciﬁcity of one of the beetles, along with the occurrence of two exclusive fungi, makes the authors suggest that the resin associated community is native and has evolved on the endemic conifer host.

line 446: ‘The coastal forest harbors a very unusual arthropod community’ – My previous question was: In what manner is it unique?

Thanks for explaining it to me, but the authors should likewise explain it to the readers.

We are included a paragraph explaining why.

Reviewer #2: 

The comments and suggestions of both reviewers have been taken into account. I feel that the manuscript has been improved and gained significantly in clarity. Therefore, I recommend the acceptance.

Although I haven’t specifically proofread the manuscript for spelling errors, I noticed a few things:

Line 57: write “detail” instead of “detain” 

Line 81: “hinders” instead of “hiders”

Line 113: “one-million-year-old” instead of “one-million-years-old”

Line 271: add a space = “between 10,000” instead of “between10,000”

Line 478: “lasts” instead of “last”

Line 533 (figure caption): add a space = “from: 1.” instead of “from:1.”

Line 587 (figure caption): replace “Montage” by “Montagne”

Line 599: italicize the entire species name

Lines 684/685: Incorrect phrasing. Did you mean: ”As explained above, we followed the proposal of Anderson (1996) suggesting a scale based on 14C dating”?

Line 737: replace “through” by “throughout”

Line 743: delete one “that” (“that lead” instead of “that that lead”)

Besides, it doesn’t look like the Table 1 is called in the text. Maybe it should be called at line 706? The sentence “The samples radiocarbon dated are the following:” is not followed by anything. A new paragraph starts after.

All the spelling errors observed by the reviewer 2 have been corrected.

---

## [Editor Report · Decision Letter 2]

20 Apr 2020

Unravelling the mystery of “Madagascar copal’: age, origin and preservation of a Recent resin

PONE-D-19-32304R2

Dear Dr. Delclos,

We are pleased to inform you that your manuscript has been judged scientifically suitable for publication and will be formally accepted for publication once it complies with all outstanding technical requirements.

With kind regards,

Phillip Barden

Academic Editor

PLOS ONE

Additional Editor Comments (optional):

Thank you for thoughtfully considering the comments in this last round. I think this will be an excellent contribution to our field!
---

## [Editor Report · Acceptance letter]

23 Apr 2020

PONE-D-19-32304R2 

Unravelling the mystery of “Madagascar copal’: age, origin and preservation of a Recent resin 

Dear Dr. Delclòs:

I am pleased to inform you that your manuscript has been deemed suitable for publication in PLOS ONE. Congratulations! Your manuscript is now with our production department. 

With kind regards,

on behalf of

Dr. Phillip Barden 

Academic Editor

PLOS ONE